# THE EMERGENCE OF REPRODUCIBILITY AND CONSISTENCY IN DIFFUSION MODELS

## ABSTRACT

Recently, diffusion models have emerged as powerful deep generative models, showcasing cutting-edge performance across various applications such as image generation, solving inverse problems, and text-to-image synthesis. These models generate new data (e.g., images) by transforming random noise inputs through a reverse diffusion process. In this work, we uncover a distinct and prevalent phenomenon within diffusion models in contrast to most other generative models, which we refer to as "consistent model reproducibility". To elaborate, our extensive experiments have consistently shown that when starting with the same initial noise input and sampling with a deterministic solver, diffusion models tend to produce nearly identical output content. This consistency holds true regardless of the choices of model architectures and training procedures. Additionally, our research has unveiled that this exceptional model reproducibility manifests in two distinct training regimes: (i) "memorization regime," characterized by a significantly overparameterized model which attains reproducibility mainly by memorizing the training data; (ii) "generalization regime," in which the model is trained on an extensive dataset, and its reproducibility emerges with the model's generalization capabilities. Our analysis provides theoretical justification for the model reproducibility in "memorization regime". Moreover, our research reveals that this valuable property generalizes to many variants of diffusion models, including conditional diffusion models, diffusion models for solving inverse problems, and fine-tuned diffusion models. A deeper understanding of this phenomenon has the potential to yield more interpretable and controllable data generative processes based on diffusion models.

## 1 INTRODUCTION

Recently, diffusion models have emerged as a powerful new family of deep generative models with remarkable performance in many applications, including image generation (Ho et al., 2020; Song et al., 2020b; Rombach et al., 2022a) , image-to-image translation (Su et al., 2022; Saharia et al., 2022; Zhao et al., 2022), text-to-image synthesis (Rombach et al., 2022a; Ramesh et al., 2021; Nichol et al., 2021), and solving inverse problem solving (Chung et al., 2022b; Song et al., 2022; Chung et al., 2022a; Song et al., 2023a). These models learn an unknown data distribution generated from the Gaussian noise distribution through a process that imitates the non-equilibrium thermodynamic diffusion process (Ho et al., 2020; Song et al., 2020b). In the forward diffusion process, the noise is continuously injected into training samples; while in the reverse diffusion process, a model is learned to remove the noise from noisy samples parametrized by a noise-predictor neural network. Then guided by the trained model, new samples (e.g., images) from the target data distribution can be generated by transforming random noise instances through step-by-step denoising following the reverse diffusion process. Despite the remarkable data generation capabilities demonstrated by diffusion models, the fundamental mechanisms driving their performance are largely under-explored.

In this work, to better understand diffusion models, we study the following fundamental question:

> **Q1:** *Starting from the **same noise input**, how are the generated data samples from various diffusion models related to each other?*

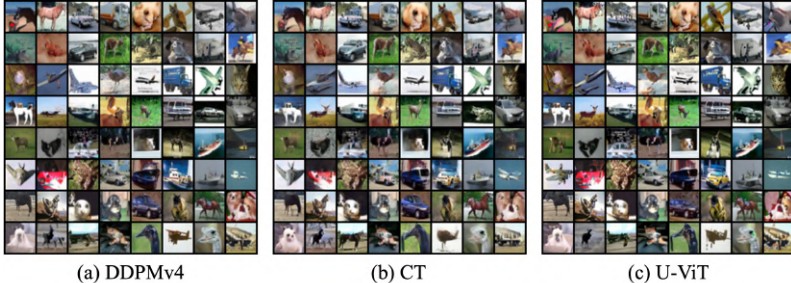

(a) DDPMv4      (b) CT      (c) U-ViT

Figure 1: **Unconditional diffusion model samples visualization in generalization regime.** We utilized denoising diffusion probabilistic models (DDPM) (Ho et al., 2020; Song et al., 2020a), consistency model (CT) (Song et al., 2023b), U-ViT (Bao et al., 2023) trained on CIFAR-10 (Krizhevsky et al., 2009) dataset released by the author. Samples in the corresponding row and column are generated from the same initial noise with a deterministic ODE sampler. Discussion of the generalization regime is in Figure 2b.

A deeper understanding of this question has the potential to yield more interpretable and controlled data generative processes of using the diffusion models in many application disciplines (Zhang et al., 2023; Epstein et al., 2023). For example, in the text-driven image generation, insights into this question could help to guide the content generation (e.g., adversial attacking (Zou et al., 2023), robust defending (Zhu et al., 2023), copyright protection (Somepalli et al., 2023b;a)) using the same text embedding but varying noise inputs. In solving inverse problems, an answer to this question will guide us to select the input noise for reducing the uncertainty and variance in our signal reconstruction (Jalal et al., 2021; Chung & Ye, 2022; Luo et al., 2023a). Theoretically, understanding the question will shed light on how the mapping function is learned and constructed between the noise and data distributions, which is crucial for understanding the generation process through the distribution transformation or identifiable encoding (Roeder et al., 2021; Khemakhem et al., 2020a;b).

In this work, we provide an in-depth study of the question and uncover an intriguing and prevalent phenomenon within the diffusion model that sets it apart from most other generative models. We term this phenomenon as "*consistent model reproducibility*". More precisely, as illustrated in Figure 1, when different diffusion models are trained on the same dataset and generate new samples through a deterministic ODE sampler from the same noises, we find that *all diffusion models generate nearly identical images, which is irrespective of network architectures, training and sampling procedures, and perturbation kernels.*

The consistent model reproducibility we identified for diffusion models is similar to the notion of unique identifiable encoding for deep latent-variable models, which is the property that the learned input-embedding is reproducible towards an identifiable mapping, regardless of different weight initialization or optimization procedures (Roeder et al., 2021). The property for deep latent-variable models was proved by Hyvarinen & Morioka (2016; 2017); Hyvarinen et al. (2019) on the analysis of Independent Component Analysis (ICA). Recently, Khemakhem et al. (2020a) demonstrated the identifiability of Variational Autoencoder (VAE) based on conditionally factorial priors distribution over the latent variables, and Roeder et al. (2021) proved a linear identifiability on representation learning. In comparison, the consistent model reproducibility of diffusion models studied in this work implies that we are learning a unique encoding between the noise space and the image space.

More interestingly, as illustrated in Figure 2, we find that the consistent model reproducibility of diffusion models emerges in two distinct regimes: (*i*) "*memorization regime*" where the model has large capacity of memorizing the training data but no ability to generate new samples, and (*ii*) "*generalization regime*" where the model regain the consistent model reproducibility and can also produce new data. In this work, we provide theoretical justification for the memorization regime. Moreover, the simultaneous occurrence of reproducibility and generalizability in the model generalization regime presents an interesting open question that is worth of future study.

Finally, we show that the consistent model reproducibility could be generalized to many other variants of diffusion model settings: conditional diffusion models, diffusion models for solving inverse problem, and fine-tuning diffusion models. Regarding conditional diffusion models, the model reproducibility is not only evident among different conditional diffusion models, but also it manifests in a structured way when comparing conditional models to their unconditional counterparts. Concerning solving inverse problem by using diffusion models as generative priors, we observe that the

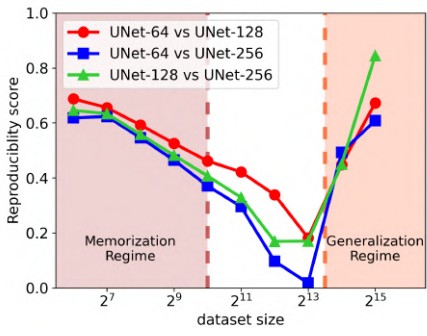
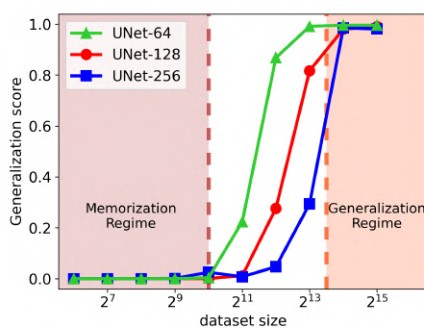

(a) Reproducibilty    (b) Generalizability

Figure 2: **"Memorization" and "Generalization" regimes for unconditional diffusion models.** We utilize DDPMv4 and train them on the CIFAR-10 dataset, adjusting both the model's size and the size of the training dataset. In terms of model size, we experiment with UNet-64, UNet-128, and UNet-256, where, for instance, UNet-64 indicates a UNet structure with an embedding dimension of 64. As for the dataset size, we select images from the CIFAR dataset, ranging from $2^6$ to $2^{15}$. Under each dataset size, different models are trained from the same subset of images. The figure on the left displays the reproducibility score as we compare various models across different dataset sizes, while the figure on the right illustrates the generalizability score of the models as the dataset size changes.

model reproducibility is confined within models using the same type of network architectures (e.g., either U-Net based or transformer-based architecture). Finally, in the context of fine-tuning diffusion models, we show partial fine-tuning of pretrained models reduces reproducibility but improves generalizability in "memorization regime" when compared to training from scratch.

**Summary of Contributions.** In summary, we briefly highlight our contributions below:

- **Model reproducibility of unconditional diffusion models (Section 2).** We provide a systematic study of the of the unconditional diffusion model, regardless of the choices of network architectures, perturbation kernels, training and sampling settings.
- **Two regimes of model reproducibility (Section 3).** We find a strong correlation between model's reproducibility and generalizability. We provide a theoretical study for the memorization regime.
- **Model reproducibility of variants of diffusion models (Section 4).** We reveal the reproducibility in more various diffusion model settings, including conditional diffusion models, diffusion models for inverse problem solving, and fine-tuning diffusion models.

Finally, we conclude and discuss the implications of our results in Section 5.

## 2 STUDY FOR UNCONDITIONAL DIFFUSION MODELS

For unconditional diffusion models, new samples (e.g., images) are generated by transforming *randomly sampled noise instances* through a reverse diffusion process guided by the trained model (Ho et al., 2020; Song et al., 2020b). If we start from the *same* noise input and use a deterministic ODE sampler, we observe that

> **C2:** *Diffusion models consistently generate **nearly identical contents**, irrespective of network architectures, training and sampling procedures, and perturbation kernels.*

This seems to be obvious by examining the visualized samples generated by different diffusion models in Figure 1 – starting from the same noise input, different diffusion models (i.e., DDPM (Ho et al., 2020), Consistency Training (CT) (Song et al., 2023b), and U-ViT (Bao et al., 2023)) generate nearly identical samples with very similar low-level color structures. More recent seminal (Song et al., 2020b) has observed a similar phenomenon (see also subsequent works (Song et al., 2023b; Karras et al., 2022)), but the study in Song et al. (2020b) remains preliminary.

### 2.1 EVALUATION SETUP OF MODEL REPRODUCIBILITY

While the findings in Figure 1 are intriguing, the basic visualization alone is not sufficient to fully justify our claim. For a more comprehensive study, we first introduce quantitative measures for model reproducibility and basic experimental setup.

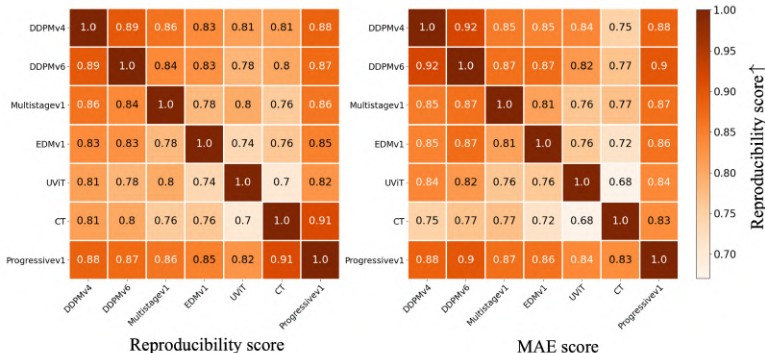

Figure 3: **Similarity among different unconditional diffusion model settings in generalization regime.** We visualize the quantitative results based upon seven different unconditional diffusion models (DDPMv4, DDPMv6 (Ho et al., 2020; Song et al., 2020a), Multistagev1 (Anonymous), EDMv1 (Karras et al., 2022), UViT (Bao et al., 2023), CT (Song et al., 2023b), Progressivev1 (Salimans & Ho, 2022)) based upon reproducibility score (left) and MAE score (right) (defined in Section 2.1). About more detailed settings and a more comprehensive compare could be found in Appendix B.

**Quantitative Measures of Model Reproducibility.** To study this phenomenon more quantitatively, we introduce the *reproducibility (RP) score* to measure the similarity of image pair generated from two different diffusion models starting from the same noise drawn i.i.d. from the standard Gaussian distribution. Specifically, we define

$$\text{RP Score} := \mathbb{P}\left(\mathcal{M}_{\text{SSCD}}(\boldsymbol{x}_1, \boldsymbol{x}_2) > 0.6\right),$$

represents the *probability* of a generated sample pair $(\boldsymbol{x}_1, \boldsymbol{x}_2)$ from two different diffusion models to have *self-supervised copy detection* (SSCD) similarity $\mathcal{M}_{\text{SSCD}}$ larger than 0.6 (Pizzi et al., 2022; Somepalli et al., 2023b) ($\mathcal{M}_{\text{SSCD}} > 0.6$ exhibits strong visual similarities). We sampled 10K noise to estimate the probability. The SSCD similarity is first introduced in Pizzi et al. (2022) to measure the replication between image pair $(\boldsymbol{x}_1, \boldsymbol{x}_2)$, which is defined as the following:

$$\mathcal{M}_{\text{SSCD}}(\boldsymbol{x}_1, \boldsymbol{x}_2) = \frac{\text{SSCD}(\boldsymbol{x}_1) \cdot \text{SSCD}(\boldsymbol{x}_2)}{||\text{SSCD}(\boldsymbol{x}_1)||_2 \cdot ||\text{SSCD}(\boldsymbol{x}_2)||_2}$$

where $\text{SSCD}(\cdot)$ represents a neural descriptor for copy detection.

In addition, we also use the *mean-absolute-error (MAE) score* to measure the reproducibility, MAE Score $:= \mathbb{P}\left(\text{MAE}(\boldsymbol{x}_1, \boldsymbol{x}_2) < 15.0\right)$, similar setting with the RP score. $\text{MAE}(\cdot)$ is the operator that measures the mean absolute different of image pairs in pixel value space ([0, 255]). The quantitative results for selected diffusion model architectures are shown in Figure 3.

**Experimental Setup for Evaluation.** In this work, we conduct a comprehensive study of model reproducibility based upon different network architectures, model perturbation kernels, and training and sampling processes: **(a) Network architectures.** We evaluate for both UNet (Ronneberger et al., 2015) based architecture: DDPM (Ho et al., 2020), DDPM++ (Song et al., 2020b), Multistage (Anonymous), EDM (Karras et al., 2022), Consistency Training (CT) and Distillation (CD) (Song et al., 2023b); Transformer (Vaswani et al., 2017) based architecture: DiT (Peebles & Xie, 2022) and U-ViT (Bao et al., 2023). **(b) Training Process.** We also considered discrete (Ho et al., 2020) or continuous (Song et al., 2020b) settings, estimating noise $\epsilon$ or original image $\boldsymbol{x}_0$, training from scratch or distillation (Salimans & Ho, 2022; Song et al., 2023b) for the diffusion model. **(c) Sampling Process.** For sampling, we only used the *deterministic* sampler,[1] such as DPM-Solver (Lu et al., 2022), Heun-Solver (Karras et al., 2022), DDIM (Song et al., 2020a) etc. **(d) Perturbation Kernels.** For noise perturbation stochastic differential equations, we use Variance Preserving (VP) (Ho et al., 2020), Variance Exploding (VE), sub Variance Preserving (sub-VP) (Song et al., 2020b). All the models are trained with the CIFAR-10 dataset (Krizhevsky et al., 2009) until convergence.[2]

---

[1]We use deterministic sampler instead of stochastic sampler, because randomness introduced by stochastic samplers will break the model reproducibility.

[2]Here, convergence means achieving the lowest Fréchet inception distance (FID) Heusel et al. (2017).

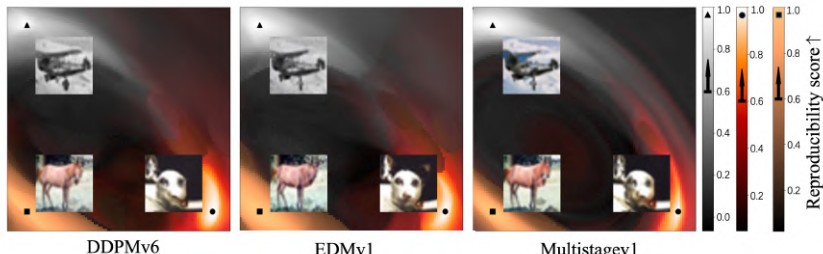

Figure 4: **Unqiue Encoding from Noise Hyperplane to Image Manifold in generalization regime.** The diagram illustrates the process of mapping from a noise hyperplane to the image manifold. We employ three distinct models: DDPMv6, EDMv1, and Multistagev1. Initially, we select three different initial noises from Gaussion and generate corresponding samples, denoted by a triangle, square, and circle in the first three images on the left. The hyperplane is defined based on these chosen noises. Image generations, starting from uniformly selected initial noise within this hyperplane, are classified as identical to either the triangle, square, or circle image, determined by the maximum SSCD similarity with them. Each initial noise is colored according to its generation's corresponding class (as indicated on the right; for instance, the noise's generation identical to the triangle image is represented by the black-white color bar), along with the SSCD similarity to the identical image.

## 2.2 MODEL REPRODUCIBILITY OF UNCONDITIONAL DIFFUSION MODELS

**Quantitative Studies of Model Reproducibility.** Based upon the setup in Section 2.1, we provide a quantitative study by comparing the similarity metrics of samples generated based upon different network architectures, model perturbation kernels, and training process. In Figure 3, we visualize the *similarity matrix* for seven different popular diffusion models, where each element of the matrix measures pairwise similarities of two different diffusion models based upon reproducibility score (left) and MAE score (right). More comprehensive studies are shown in Appendix B. As we can see, there is a very consistent model reproducible phenomenon for comparing any two models. For even the most dissimilar models, (i.e., CT and UViT), the RP and MAE scores are notably high at 0.7 and 0.64, respectively. Specifically, we draw the following conclusions:

- *Model reproducibility is consistent among different network architectures.* When compared to DDPMv4, Multistagev1, EDMv1, and UViT, this phenomenon remains consistent regardless of the specific architecture employed.

- *Model reproducibility is consistent among different training procedures.* When we compare CT (consistency loss) and EDMv1 (diffusion model loss), even when using different loss functions, they ultimately converge to a similar noise-to-image mapping. Notably, EDMv1 employs a distinct loss by estimating the clear image $x$ whereas other methods estimate the noise $\epsilon$. Additionally, comparing DDPMv1 and Progressivev1 reveals that both training from scratch and distillation approaches lead to the same results.

- *Model reproducibility is consistent among different sampling procedures.* DDPMv4 utilizes DPM-solver, EDMv1 employs a 2nd order heun-solver, and CT utilizes consistency sampling, yet they all exhibit reproducibility.

- *Model reproducibility is consistent among different perturbation kernels.* When compared to DDPMv4, DDPMv6, and EDMv1, the reproducibility remains unaffected by the choice of perturbation method (VP, sub-VP, and VE, respectively).

**Studies of the Unqiue Encoding from Noise Hyperplane to Image Manifold.** The model reproducibility of diffusion models implies that we are learning a unique decoding $f : \mathcal{E} \mapsto \mathcal{I}$ from the gaussian noise space $\mathcal{E}$ to the image manifold $\mathcal{I}$. Specifically, we find that

- *Similar unique encoding maps across different network architectures.* We further confirm the model reproducibility by visualizing the mapping $f$ from a 2D noise hyperplane $\mathcal{H} \subseteq \mathcal{E}$ to the image manifold $\mathcal{I}$, inspired by Somepalli et al. (2022). The visualization in Figure 4 shows that different mappings of different network architectures share very similar structures.

- *Local Lipschitzness of the unique encoding from noise to image space.* Furthermore, our visualization suggests that the unique encoding $f$ is locally Lipschitz, where $\|f(\epsilon_1)-f(\epsilon_2)\| \leq L\|\epsilon_1-\epsilon_2\|$ for any $\epsilon_1, \epsilon_2 \in \mathcal{B}(\epsilon, \delta) \cap \mathcal{E}$ with some Lipschitz constant $L$. Here $\mathcal{B}(\epsilon, \delta)$ denotes a ball centered at a Gaussian noise $\epsilon$ with radius $\delta$. In other words, noises $\epsilon_1, \epsilon_2 \in \mathcal{E}$ close in distance would generate similar reproducible images in $\mathcal{I}$ via diffusion models.

Specifically, the visualization in Figure 4 is created as follows. First, we pick three initial noises $(\boldsymbol{\epsilon}_1, \boldsymbol{\epsilon}_2, \boldsymbol{\epsilon}_3)$ in the noise space $\mathcal{E}$ and used different diffusion model architectures to generate clear images $(\boldsymbol{x}_1, \boldsymbol{x}_2, \boldsymbol{x}_3)$ in the image manifold $\mathcal{I}$, so that the images $\{\boldsymbol{x}_i\}_{i=1}^3$ belong to three different classes. Second, we create a 2D noise hyperplane with

$$\boldsymbol{\epsilon}(\alpha, \beta) = \alpha \cdot (\boldsymbol{\epsilon}_2 - \boldsymbol{\epsilon}_1) + \beta \cdot (\boldsymbol{\epsilon}_3 - \boldsymbol{\epsilon}_1) + \boldsymbol{\epsilon}_1$$

Within the region $(\alpha, \beta) \in [-0.1, 1.1] \times [-0.1, 1, 1]$, we uniformly sample 100 points along each axis and generate images $\boldsymbol{x}(\alpha, \beta)$ for each sample $\boldsymbol{\epsilon}(\alpha, \beta)$ using different diffusion model architectures (i.e., DDPMv6, EDMv1, Multistagev1). For each point $(\alpha, \beta)$, it is considered as identical to image $\boldsymbol{x}_i$ for $i = \arg\max_{k \in \{1,2,3\}}[\mathcal{M}_{\text{SSCD}}(\boldsymbol{x}_k, \boldsymbol{x}(\alpha, \beta))]$, and we visualize the value of $\mathcal{M}_{\text{SSCD}}(\boldsymbol{x}_i, \boldsymbol{x}(\alpha, \beta))$. As we observe from Figure 4, the visualization shares very similar structures across different network architectures. Second, for each plot, closeby noises create images with very high similarities. These observations support our above claims.

**Comparison with Other Types of Generative Models.** We end this section by highlighting that only diffusion models appear to consistently exhibit model reproducibility. This property is seemingly absent in other generative models with one exception as noted in Khemakhem et al. (2020a) [3]. Details reproducibility analysis of Generative Adversarial Network (GAN) Goodfellow et al. (2014) and Variational Autoencoder (VAE) Kingma & Welling (2013) based approaches are in Appendix C.

## 3 CORRELATION BETWEEN REPRODUCIBILITY & GENERALIZABILITY

More interestingly, both of our empirical and theoretical studies in this section demonstrate that

> **C3:** *The consistent model reproducibility of diffusion models manifests in two distinct training regimes, both **strongly correlated** with the model's generalizability.*

Here, the model's generalizability means the model's ability to generate new samples that is different from samples in the training dataset. Specifically, we measure the generalizability by introducing the *generalization (GL) score* $:= 1 - \mathbb{P}\left(\max_{i \in [N]}[\mathcal{M}_{\text{SSCD}}(\boldsymbol{x}, \boldsymbol{y}_i)] > 0.6\right)$ represents one minus the *probability* of maximum $\mathcal{M}_{\text{SSCD}}$ is larger than $0.6$, between the generated sample $\boldsymbol{x}$ from one diffusion model and all samples $\boldsymbol{y}_i$ from training dataset $\{\boldsymbol{y}_i\}_{i=1}^N$ of $N$-samples. We sampled 10K initial noise to estimate the probability.

**Two Regimes of Model Reproducibility.** Now, we formally introduce the two regimes of model reproducibility for diffusion models in the following. Both of them exhibit a strong correlation with the model's ability to generalize.

- **"Memorization regime"** characterizes a scenario where the trained model has large model capacity than the size of training data. In this regime, starting from the same noise the model possesses the ability to reproduce the same results, as illustrated in the left region of Figure 2a with small training data size. However, the generated samples are often replications of the samples in the training data, and the model lacks the full capacity to generate new samples; see the left region of Figure 2b. Therefore, we call this regime the "memorization regime" as the generated samples are only replication of training data. In this regime, we can rigorously characterize the optimal denoiser as shown in Theorem 1, and in Figure 5 we empirically demonstrated that the diffusion models converge to the theoretical solutions when the model is highly overparameterized. However, given no generalizability, training diffusion models in this regime holds limited practical interest.

- **"Generalization regime"** emerges when the diffusion model is trained on large dataset and doesn't have full capacity to memorize the whole dataset. Remarkably, as shown in the right region of Figure 2a, the diffusion model regains the reproducibility as the ratio between dataset size and model capacity increases. Simultaneously, the model possess the ability to generate new samples, as illustrated in the right region of Figure 2b; this generalization phenomenon is also observed in Yoon et al. (2023). As such, we call this regime the "generalization regime" because the stage of generalization is in coincidence with the stage of model reproducibility. This is the regime in which diffusion models are commonly trained and employed in practice, and the coexistence of model reproducibility and generalizability is an intriguing phenomenon to be further understood theoretically.

---

[3]Khemakhem et al. (2020a) demonstrates that VAE is uniquely identifiable encoding given a factorized prior distribution over the latent variables.

**A Theoretical Study of the Memorization Regime.** For the rest of this section, we provide a theoretical study of the memorization regime, and we leave the study of generalization regime for future work. Specifically, we characterize the optimal denoiser of the diffusion model and show that it results in a unique identifiable encoding under proper assumptions.

**Theorem 1.** *Suppose we train a diffusion model denoiser function $\epsilon_{\boldsymbol{\theta}}(\boldsymbol{x}, t)$ with parameter $\boldsymbol{\theta}$ on a training dataset $\{\boldsymbol{y}_i\}_{i=1}^N$ of $N$-samples, by minimizing the training loss*

$$\min_{\boldsymbol{\theta}} \mathcal{L}(\epsilon_{\boldsymbol{\theta}}; t) = \mathbb{E}_{\boldsymbol{x}_0 \sim p_{data}(\boldsymbol{x})} \mathbb{E}_{\boldsymbol{x} \sim p_t(\boldsymbol{x}|\boldsymbol{x}_0)} [||\epsilon - \epsilon_{\boldsymbol{\theta}}(\boldsymbol{x}, t)||^2], \tag{1}$$

*where we assume that data $\boldsymbol{x}_0$ follows a mixture delta distribution $p_{data}(\boldsymbol{x}) = \frac{1}{N} \sum_{i=1}^N \delta(\boldsymbol{x} - \boldsymbol{y}_i)$, and the perturbation kernel $p_t(\boldsymbol{x}_t|\boldsymbol{x}_0) = \mathcal{N}(\boldsymbol{x}_t; s_t \boldsymbol{x}_0, s_t^2 \sigma_t^2 \boldsymbol{I})$ with perturbation parameters $s_t, \sigma_t$. Then we can show that the optimal denoiser $\epsilon_{\boldsymbol{\theta}}^*(\boldsymbol{x}; t) = \arg\min_{\epsilon_{\boldsymbol{\theta}}} \mathcal{L}(\epsilon_{\boldsymbol{\theta}}; t)$ is*

$$\epsilon_{\boldsymbol{\theta}}^*(\boldsymbol{x}; t) = \frac{1}{s_t \sigma_t} \left[ \boldsymbol{x} - s_t \frac{\sum_{i=1}^N \mathcal{N}(\boldsymbol{x}; s_t \boldsymbol{y}_i, s_t^2 \sigma_t^2 \boldsymbol{I}) \boldsymbol{y}_i}{\sum_{i=1}^N \mathcal{N}(\boldsymbol{x}; s_t \boldsymbol{y}_i, s_t^2 \sigma_t^2 \boldsymbol{I})} \right]. \tag{2}$$

*Moreover, suppose a trained diffusion model could converge to the optimal denoiser $\epsilon_{\boldsymbol{\theta}}^*(\boldsymbol{x}; t)$ and we use a deterministic ODE sampler to generate images using $\epsilon_{\boldsymbol{\theta}}^*(\boldsymbol{x}; t)$, then $f : \mathcal{E} \mapsto \mathcal{I}$, which is determined by the $\epsilon_{\boldsymbol{\theta}}^*(\boldsymbol{x}; t)$ and the ODE sampler, is an invertiable mapping and the inverse mapping $f^{-1}$ is a unique identifiable encoding.*

The proof for Theorem 1 can be found in the Appendix D, building upon previous findings from Karras et al. (2022); Yi et al. (2023). It is worth noting that the optimal denoiser $\epsilon_{\boldsymbol{\theta}}^*(\boldsymbol{x}; t)$ in (2) is deterministic for a given training dataset $\{\boldsymbol{y}_i\}_{i=1}^N$ and for specific perturbation parameters $s_t, \sigma_t$. Moreover, Yi et al. (2023) demonstrates optimal denoiser lacks generalizability, as all the samples generated from $\epsilon_{\boldsymbol{\theta}}^*(\boldsymbol{x}; t)$ are confined within the training dataset. Furthermore, we verified our theory experimentally by comparing the reproducibility score between the the theoretical generations in Theorem 1 and that of a trained network in practice. This can be illustrated by the left region of Figure 5, the trained networks have a very high similarity compared with the theoretical solution on small training dataset, and the similarity decreases when the size of training data increases. As such, we conclude that the diffusion model could converge to the theoretical solution when the model capacity is large enough.

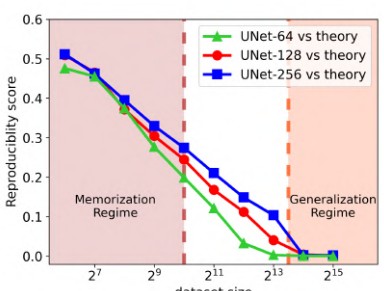

Figure 5: **Experiment verification of the theory.** We employ DDPMv4 and conduct training on the CIFAR-10 dataset. During this process, we make modifications to both the model's capacity and the size of the training dataset, maintaining the same configuration as depicted in Figure 2. The figure presented here illustrates the reproducibility score between each diffusion model and the theoretically unique identifiable encoding as outlined in Theorem 1.

## 4    STUDY BEYOND UNCONDITIONAL DIFFUSION MODELS

Furthermore, in this section, we explore how model reproducibility can be generalized to variations of the diffusion models, showcasing that:

> **C4:** *Model reproducibility holds **more generally** across conditional diffusion models, diffusion models for inverse problems, the fine-tuning of diffusion models.*

**Model Reproducibility in Conditional Diffusion Models.** Conditional diffusion, introduced by Ho & Salimans (2022); Dhariwal & Nichol (2021), gained its popularity in many applications such as text-to-image generation (Rombach et al., 2022a; Ramesh et al., 2021; Nichol et al., 2021) with superior performance, through the fusion of rich class embeddings with the denoiser function. Interestingly, we find that:

*Model reproducibility of conditional models is evident and linked with unconditional counterparts.*

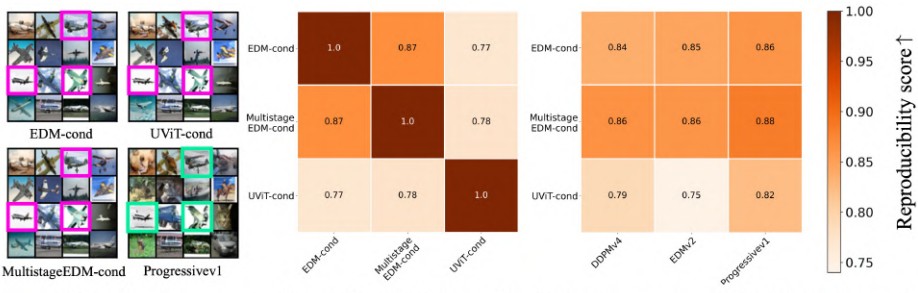

(a) Visualization     (b) Conditional diffusion models    (c) Conditional and unconditional diffusion models

Figure 6: **Model reproducibility for conditional diffusion model in the generalization regime.** In this study, we employ conditional diffusion models, specifically U-Net-based (EDM-cond, MultistageEDM-cond) and transformer based (UViT-cond), which we train on the CIFAR-10 dataset using class labels as conditions. Additionally, we select unconditional diffusion models, namely Progressivev1, DDPMv4, and EDMv2, as introduced in Section 2. Figure (a) showcases sample generations from both unconditional and conditional diffusion models (with the "plane" serving as the condition for the latter). Notably, samples within the same row and column originate from the same initial noise. The reproducibility scores between the conditional diffusion models are presented in (b), and between unconditional and conditional diffusion models in (c).

Specifically, our experiments in Figure 6 demonstrate that (*i*) model reproducibility exists among different conditional diffusion models, and (*ii*) model reproducibility is present between conditional and unconditional diffusion models *only* if the type (or class) of content generated by the unconditional models matches that of the conditional models. More results can be found in Appendix E.

To support our claims, we introduce the *conditional reproducibility score* between different conditional diffusion models by $\text{RP}_{cond}$ Score $:= \mathbb{P}\left(\mathcal{M}_{\text{SSCD}}(\boldsymbol{x}_1^c, \boldsymbol{x}_2^c) > 0.6 \mid c \in \mathcal{C}\right)$, where the pair $(\boldsymbol{x}_1^c, \boldsymbol{x}_2^c)$ are generated by two conditional models from the same initial noise and conditioned on the class $c \in \mathcal{C}$. Additionally, the *between reproducibility score* of conditional and unconditional diffusion models is defined as $\text{RP}_{between}$ Score $:= \mathbb{P}\left(\max_{c \in \mathcal{C}}[\mathcal{M}_{\text{SSCD}}(\boldsymbol{x}_1, \boldsymbol{x}_2^c)] > 0.6\right)$, for an unconditional generation $\boldsymbol{x}_1$ and conditional generation $\boldsymbol{x}_2^c$ starting from the same noise. First, when conditioned on the same class and initial noise, results in Figure 6 (a) (b) highlight the similarity of samples generated from different condition diffusion models (i.e., EDM-cond, UViT-cond, and MultistageEDM-cond), supporting Claim (*i*). Second, the high $\text{RP}_{between}$ Score in Figure 6 (c) provides strong evidence for reproducibility between conditional and unconditional diffusion models, supporting Claim (*ii*). This can also be observed visually by examining the unconditional generation using Progressivev1 in Figure 6 (a), where images highlighted in the green square share high similarity with conditional counterparts highlighted in the purple square .

**Model Reproducibility in Solving Inverse Problems.** Recently, diffusion models have also demonstrated remarkable results on solving a broad spectrum of inverse problems (Song et al., 2023a; Chung et al., 2022a; Song et al., 2021; Chung et al., 2022b),[4] including but not limited to image super-resolution, de-blurring, and inpainting. Motivated by these promising results, we illustrate based upon solving the image inpainting problem using a modified deterministic variant of diffusion posterior sampling (DPS) (Chung et al., 2022a), showcasing that for solving inverse problem using diffusion models:

*Model reproducibility largely holds only within the same type of network architectures.*

Our claim is supported by the experimental results in Figure 7. Specifically, Figure 7 (a) virtualizes the samples generated from different diffusion models, and Figure 7 (b) presents the similarity matrix of model reproducibility between different models, i.e., U-Net based (DDPMv1, DDPMv2, DDPMv3, DDPMv4, Multistagev1) and Transformer based (DiT, U-ViT) architectures. We note a strong degree of model reproducibility *among* architectures of the same type (such as U-Net or Transformer), but the model reproducibility score exhibits a notable decrease when comparing any U-Net model to any Transformer-based model.

**Model Reproducibility in Fine-tuning Diffusion Models.** Few-shot image fine-tuning for diffusion models, as discussed in (Ruiz et al., 2023; Gal et al., 2022; Moon et al., 2022; Han et al., 2023), showcases remarkable generalizability. This is often achieved by fine-tuning a small portion of the

---

[4]Here, the problem is often to reconstruct an unknown signal $\boldsymbol{u}$ from the measurements $\boldsymbol{z}$ of the form $\boldsymbol{z} = \mathcal{A}(\boldsymbol{u}) + \boldsymbol{\eta}$, where $\mathcal{A}$ denotes some (given) sensing operator and $\boldsymbol{\eta}$ is the noise.

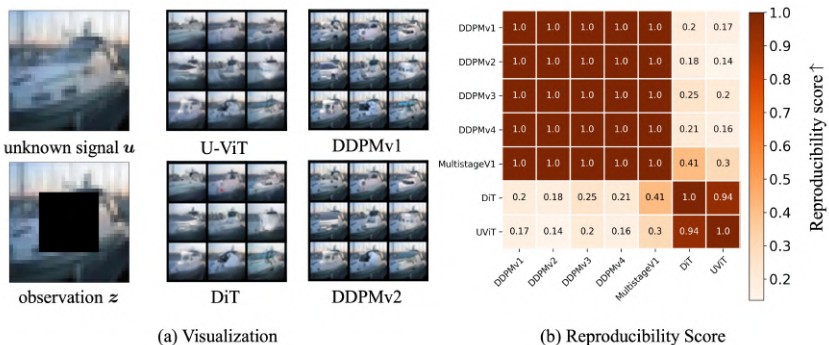

Figure 7: **Model reproducibility for solving inverse problems in the generalization regime.** In this investigation, we employ various unconditional diffusion models, as introduced in Section 2, which were initially trained on the CIFAR-10 dataset. Our approach involves utilizing a modified deterministic variant of diffusion posterior sampling (DPS), as detailed in Appendix F. Specifically, we focus on the task of image inpainting. Figure (a) presents both the observation $z$, unknown signal $u$, and generations from different diffusion models. Notably, samples within the same row and column originate from the same initial noise. The reproducibility scores for different diffusion models under the DPS algorithm are quantitatively analyzed in (b).

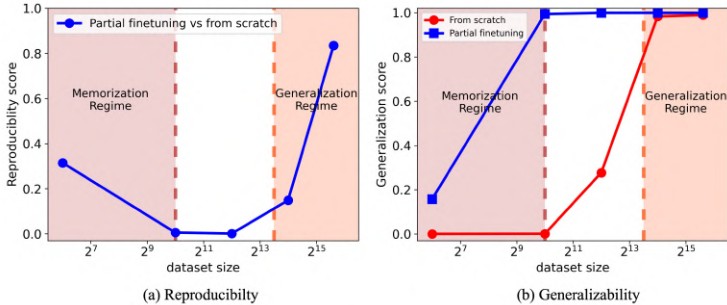

Figure 8: **Model reproducibility for diffusion model finetuing.** In this experiment, we employ DDPMv4. Two distinct training strategies are investigated: "from scratch," denoting direct training on a subset of the CIFAR-10 dataset, and "partial fine-tuning," which involves pretraining on the entire CIFAR-100 dataset Krizhevsky et al. (2009) followed by fine-tuning only the attention layers of the model on a subset of the CIFAR-10 dataset. The dataset sizes for CIFAR-10 range from $2^6$ to $2^{15}$. Importantly, both "from scratch" and "partial fine-tuning" are trained using the same subset of images for each dataset size. Under different dataset sieze, Figure (a) illustrates the reproducibility score between these two strategies and (b) presents the generalization score for them.

parameters of a large-scale pre-trained (text-to-image) diffusion model. In this final study, we delve into the impacts of partial model fine-tuning on both model reproducibility and generalizability, by extending our analysis in Section 3. We show that:

*Partial fine-tuning reduces reproducibility but improves generalizability in "memorization regime".*

Our claim is supported our results in Figure 8, comparing model fine-tuning and training from scratch of with varying size of the training data, where both models have the same number of parameters. In comparison to training from scratch that we studied in Figure 2b, fine-tuning specific components of pre-trained diffusion models, particularly the attention layer in the U-Net architecture, yields lower model reproducibility score but higher generalization score in the memorization regime. However, in the generalization regime, partial model fine-tuning has a minor impact on both reproducibility and generalization in the diffusion model. Our result reconfirms the improved generalizability of fine-tuning diffusion models on limited data, but shows a surprising tradeoff in terms of model reproducibility that is worth of further investigations.

## 5 CONCLUSION

In this work, we conducted an in-depth study of an important but largely overlooked phenomenon in diffusion models, for which we term it as "consistent model reproducibility". This study raises numerous compelling questions that is worth of further exploration. One such question is the tangible practical benefits of model reproducibility in diffusion models compared to other types of generative models. Moreover, the strong connection between model reproducibility and generalizability opens an enticing theoretical question for further study.

# Reproducibility Statement

To illustrate the intriguing phenomenon of consistent model reproducibility, there is no requirement to provide any code or model for validation. The only thing provided is the initial random noise we used for the 8×8 image grid. By selecting one diffusion mode list in the paper (you could also explore other diffusion models not listed in the paper), you can access their released model online. The only prerequisite is that the training dataset should be CIFAR-10. By following this approach, you could regenerate the samples for unconditional in Figure 1,16 and for conditional in Figure 20, 21. The details of the experiment for the unconditional model are in B, for the conditional diffusion model are in E, for theoretical verification are in D, for diffusion models on inverse problems solving are in F and for fine-tuning diffusion models are in G.

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

# Appendix

We include more comprehensive experiment settings, quantitative results, and detailed discussion of the unconditional diffusion model in Appendix B, theoretical analysis on "memorization regime" in Appendix D, conditional diffusion model in Appendix E, diffusion model for solving inverse problems in Appendix F, fine-tuning diffusion model in Appendix G. Quantitative analysis of other generative models is in Appendix C. Proof for Theorem 1 could also be found in Appendix D.

## A EXTRA EXPERIMENTS

### A.1 EXPERIMENTS ON IMAGENET DATASET

In addition to exploring the CIFAR-10 dataset, our study extends to evaluate the reproducibility of conditional diffusion models on the ImageNet dataset (Deng et al., 2009). Specifically, we focus on two models: EDM Karras et al. (2022) and ADM (Dhariwal & Nichol, 2021). For this experiment, we generate 10k initial noise paired with random class labels. The reproducibility score calculation remains largely consistent with the methodology outlined in Section 4. However, we adjusted the threshold for SSCD similarity from 0.6 to 0.4. This modification accounts for the increased complexity in resolution and semantics of the ImageNet dataset and aligns with the threshold used in the original study by (Somepalli et al., 2023b).

Our findings, depicted in Figure 9c, indicate that approximately 81% of the generated images from these two models exhibit an SSCD similarity exceeding the 0.4 threshold. Furthermore, as illustrated in Figures 9a and 9b, there is a notable visual similarity between the generations from both models.

### A.2 TEXT-TO-IMAGE DIFFUSION MODEL

Our study also explores the reproducibility of the text-to-image diffusion model, Stable Diffusion Rombach et al. (2022a), trained on the LAION-5B dataset Schuhmann et al. (2022). We utilize the series of pre-trained Stable Diffusion models (versions v1-1 to v1-4) released by Rombach et al. (2022b). These models exhibit key differences:

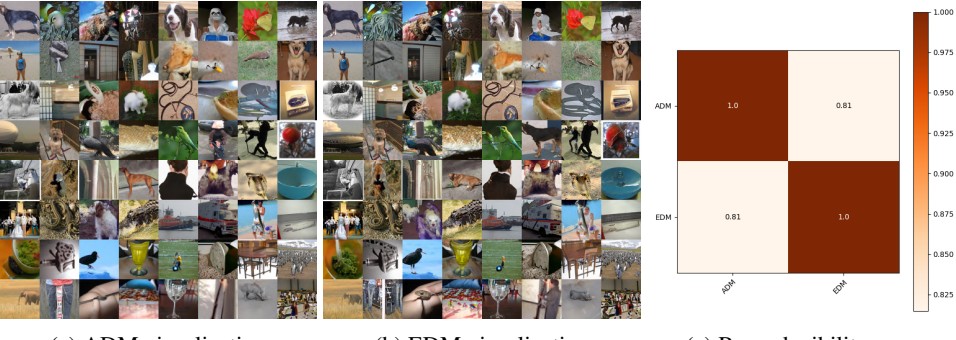

(a) ADM visualization      (b) EDM visualization      (c) Reproducibility score

Figure 9: **Reproducibility of conditional diffusion model generations on ImageNet dataset.**

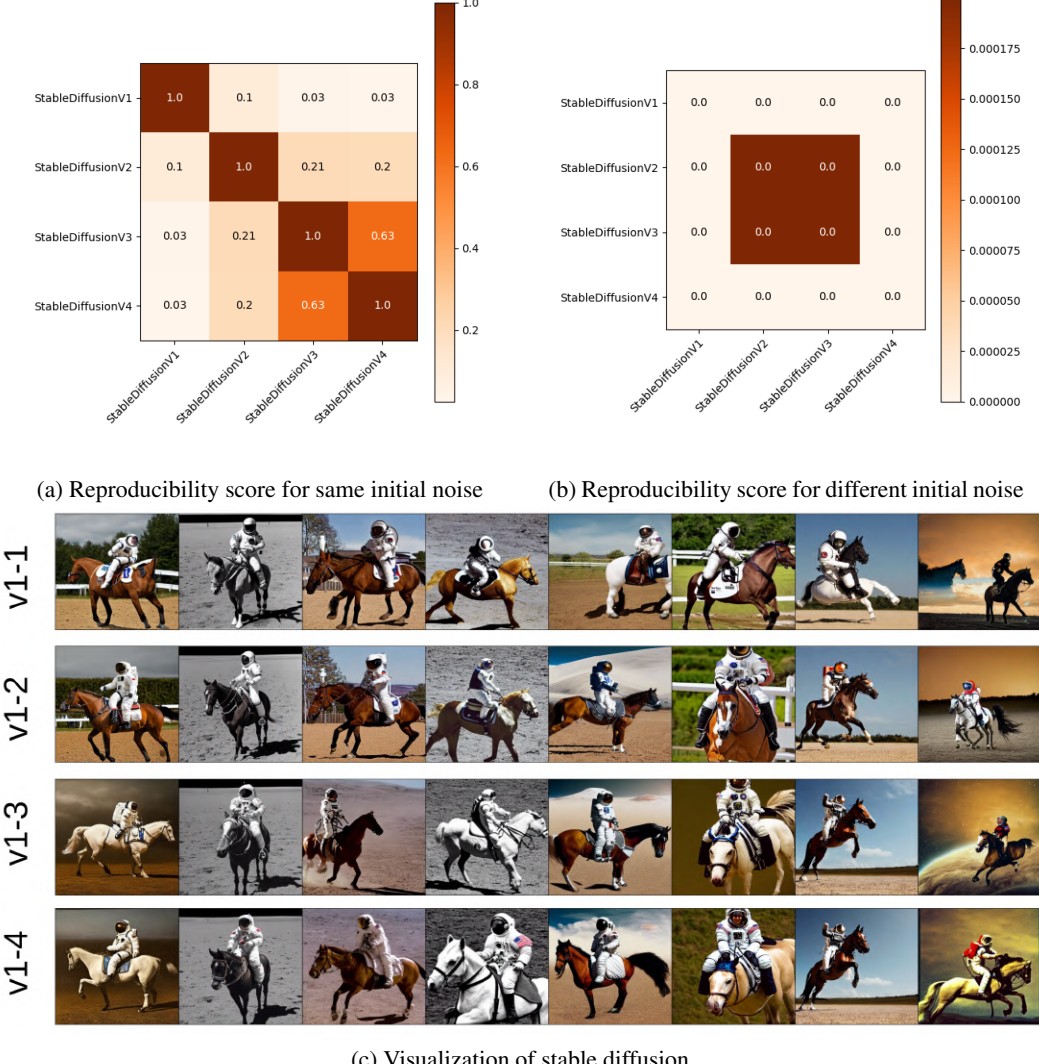

(a) Reproducibility score for same initial noise      (b) Reproducibility score for different initial noise

(c) Visualization of stable diffusion.

Figure 10: **Reproducibility of Stable Diffusion.**

- Versions v1-1, v1-2, and v1-3 each are trained on different subsets of the LAION-5B dataset.

- Versions v1-3 and v1-4 share the same training subset from LAION-5B.

- Version v1-2 is resumed from v1-1, while v1-3 and v1-4 are resumed from v1-2.

Further details on their training settings are available at Rombach et al. (2022b).

For reproducibility assessment, we use the prompt "a photograph of an astronaut riding a horse" along with 1,000 randomly generated initial noises. The reproducibility score is determined the same as the one in Section A.1. To isolate the impact of the guiding prompt on reproducibility, we also evaluate the reproducibility score with the same prompt but different initial noises.

The results, shown in Figure 10a, reveal the highest reproducibility score between v1-3 and v1-4 (0.63), likely due to their same training datasets. Lesser but noticeable reproducibility scores (below 0.21) are observed among v1-1, v1-2, and v1-3, which might be attributable to their sequential training and overlapping datasets. This finding aligns with Kadkhodaie et al. (2023), suggesting that training on exclusive subsets of the same dataset can yield reproducible results in diffusion models. A notable observation in Figure 10c is the presence of flip generations between v1-3 and v1-4, potentially a result of data augmentation introducing randomness. We hypothesize that excluding data augmentation could further increase the reproducibility score between v1-3 and v1-4. Furthermore, when varying the initial noise but with the same prompt, the reproducibility scores approach zero, as evidenced in Figure 10b, indicating only the same prompt but different initial noise will not have reproducibility.

## A.3 MANIFOLD REPRODUCIBILITY ACROSS UNCONDITIONAL DIFFUSION MODELS

This section delves into manifold reproducibility across various unconditional diffusion models, complementing the visualizations in Figure 4. We employ spherical linear interpolation (slerp) Shoemake (1985); Song et al. (2020a) to maintain approximate uniform probability distribution across all interpolation points. The process begins by selecting two initial noise vectors, $(\boldsymbol{\epsilon}^{(0)}, \boldsymbol{\epsilon}^{(1)})$, from the noise space $\mathcal{E}$. These vectors are then processed through two distinct diffusion model architectures, resulting in pairs of clear images: $(\boldsymbol{x}_1^{(0)}, \boldsymbol{x}_1^{(1)})$ from the first model and $(\boldsymbol{x}_2^{(0)}, \boldsymbol{x}_2^{(1)})$ from the second.

The manifold reproducibility score for these two models is defined as follows:

$$\text{RP}_{manifold} \text{ Score} := \mathbb{P}\left(\mathcal{M}_{\text{SSCD}}(\boldsymbol{x}_1^{(\alpha)}, \boldsymbol{x}_2^{(\alpha)}) > 0.6 \mid \alpha \in [0, 1]\right)$$

where $\boldsymbol{x}_i^{(\alpha)}$ are generated from spherical linear interpolation based on $(\boldsymbol{x}_i^{(0)}, \boldsymbol{x}_i^{(1)})$:

$$\boldsymbol{x}_i^{(\alpha)} = \frac{\sin((1-\alpha)\theta)}{\sin(\theta)}\boldsymbol{x}_i^{(0)} + \frac{\sin(\alpha\theta)}{\theta}\boldsymbol{x}_i^{(1)}, \ \ i \in \{1, 2\}$$

$$\theta = \arccos\left(\frac{\left(\boldsymbol{x}_i^{(0)}\right)^T \boldsymbol{x}_i^{(1)}}{||\boldsymbol{x}_i^{(0)}||||\boldsymbol{x}_i^{(1)}||}\right)$$

Figure 11 showcases the manifold reproducibility scores for models such as ddpmv6, EDMv1, and Multistagev1. It's important to distinguish between the RP Score discussed in Section 2.1 and the $\text{RP}_{manifold}$ Score. While the former assesses global reproducibility on a sparser initial noise spectrum, the latter focuses on local reproducibility.

## A.4 REPRODUCIBILITY WITH RESPECT TO NUMBER OF FUNCTION EVALUATION (NFE)

This subsection investigates how the Number of Function Evaluations (NFE) influences model reproducibility, utilizing the EDMv1 model (VP SDE and Heun-Solver). Our findings are visually presented in Figure 12, where the NFE for the ODE sampler ranges from 9 to 159. Notably, while a lower NFE tends to degrade the generation quality, our observations reveal that the content of the generated images remains remarkably consistent across varying NFE levels.

## A.5 TRAINING LOSS IN MEMORIZATION AND GENERALIZATION REGIMES

This subsection focuses on evaluating the training loss of diffusion models, as defined by Equation 1 in Theorem 1. We analyze models with varying capacities (U-Net 64, 128, 256) trained on datasets

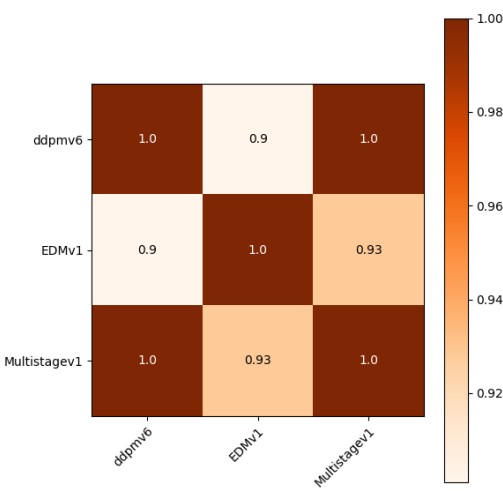

Figure 11: **Manifold reproducibility score.**

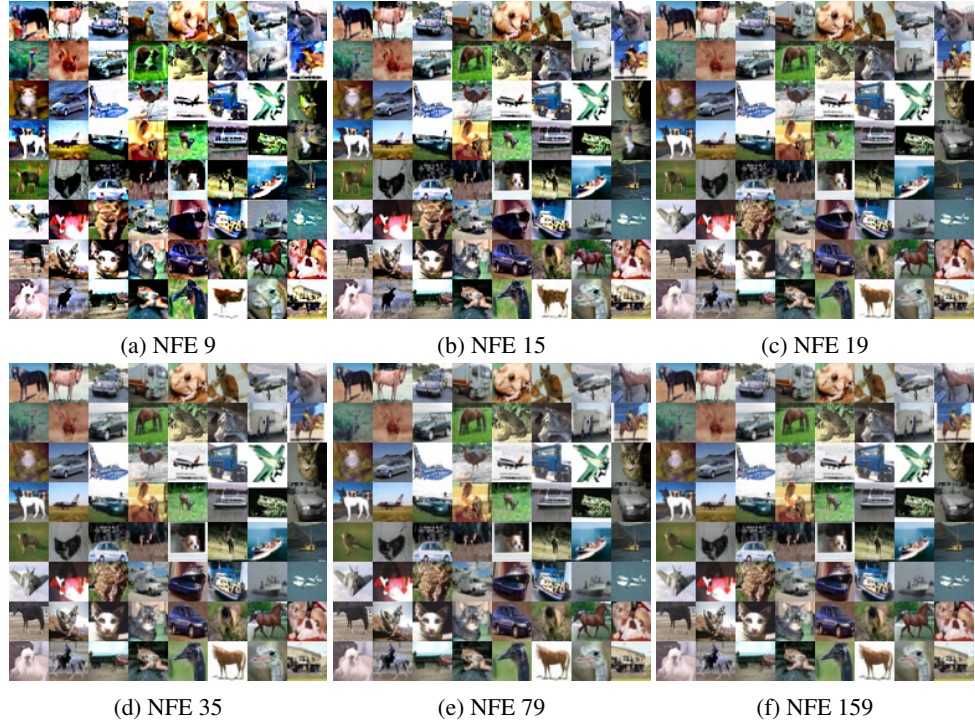

| (a) NFE 9 | (b) NFE 15 | (c) NFE 19 |
| (d) NFE 35 | (e) NFE 79 | (f) NFE 159 |

Figure 12: **Reproducibility with respect to number of function evaluation (NFE)**

of different sizes, with results illustrated in Figure 13. Notably, comparing Figure 13 with Figure 5, a lower training loss indicates a closer alignment of the trained diffusion models with the optimal denoiser. This correlation supports our theorem that the optimal denoiser is the denoiser that minimizes the training loss. This further indicates that in the memorization region, with a limited dataset and sufficient model capacity, the diffusion model effectively fits the training objective, (with a low training loss), and converges towards the optimal denoiser, so reproducibility in this stage is well studied. Conversely, in scenarios where data samples are plentiful but the model capacity is inadequate, the model struggles to fit the training data, leading to increased training loss and deviation from the optimal denoiser. However, it's noteworthy that in this latter scenario, despite the less-than-perfect fit, the model begins to demonstrate generalization capabilities while still maintaining reproducibility.

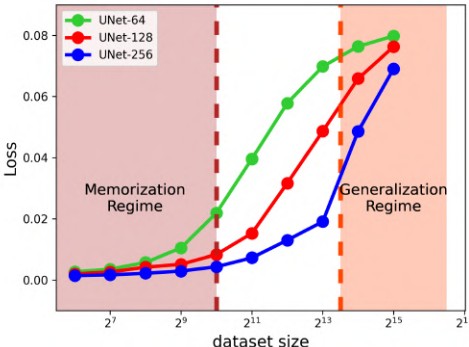

Figure 13: **Training loss for memorization and generalization regimes.**

# B UNCONDITIONAL DIFFUSION MODEL

**Expanded experiment setting** More detailed settings of the diffusion model we selected are listed in Table 1. With the exception of DiT and UViT, where we implemented and trained them ourselves, all selected diffusion model architectures utilize the author-released models.

**Architectural Relationships** For DDPMv1, DDPMv2, and DDPMv7, we adopt the DDPM architecture initially proposed by Ho et al. (2020), but we implement it using the codebase provided by Song et al. (2020b). DDPMv3 and DDPMv8, on the other hand, employ DDPM++, an enhanced version of DDPM introduced by Song et al. (2020b). DDPM++ incorporates BigGAN-style upsampling and downsampling techniques, following the work of Brock et al. (2018). DDPMv4, DDPMv5, and DDPMv6 adopt DDPM++(deep), which shares similarities with DDPM++ but boasts a greater number of network parameters. Moving to Multistagev1, Multistagev2, and Multistagev3, these models derive from the Multistage architecture, a variant of the U-Net architecture found in DDPM++(deep). For EDMv1, EDMv2, CT, and CD, the EDM architecture is identical to DDPM++, but they differ in their training parameterizations compared to other DDPM++-based architectures. Finally, UViT and DiT are transformer-based architectures.

**Distillation Relationships** CD, Progressivev1, Progressivev2, and Progressivev3 are all diffusion models trained using distillation techniques. CD employs EDM as its teacher model, while Progressivev1, Progressivev2, and Progressivev3 share DDPMv3 as their teacher model. It's worth noting that these models employ a progressive distillation strategy, with slight variations in their respective teacher models, as elaborated in Salimans & Ho (2022).

**Initial Noise Consistency** However, it is important to note a nuanced difference related to the noise perturbation kernels. Specifically, for VP and subVP noise perturbation kernels, we define the noise space as $\mathcal{E} = \mathcal{N}(\mathbf{0}, \mathbf{I})$, whereas the VE noise perturbation kernel introduces a distinct noise space with $\mathcal{E} = \mathcal{N}(\mathbf{0}, \sigma_{\max}^2 \cdot I)$, where $\sigma_{\max}$ is predefined. So during the experiment, we sample 10K initial noise $\epsilon_{\text{vp, subvp}} \sim \mathcal{N}(\mathbf{0}, \mathbf{I})$ for the sample generation of diffusion models with VP and subVP noise perturbation kernel. For diffusion models with VE noise perturbation kernel, the initial noise is scaled as $\epsilon_{\text{ve}} = \sigma_{\max} \epsilon_{\text{vp, subvp}}$.

Additionally, it's worth mentioning that for all 8x8 image grids shown in the Figure 1, 16, 18, 19, 20, 21, 23, 24, 26 no matter for the unconditional diffusion model, conditional diffusion model, diffusion model for the inverse problem, or fine-tuning diffusion model, we consistently employ the same 8x8 initial noise configuration. The same setting applies to 10k initial noises for reproducibility score. This specific design is for more consistent results between different variants of diffusion models (e.g., we could clearly find the relationship between the unconditional diffusion model and conditional diffusion model by comparing Figure 16 and Figure 20, 21).

**Further discussion** In Figure 16, we provide additional visualizations, offering a more comprehensive perspective on our findings. For a deeper understanding of our results, we present extensive quantitative data in Figure 15 and Figure 14. Building upon the conclusions drawn in Section 2, we delve into the consistency of model reproducibility across discrete and continuous timestep set-

Table 1: **Comprehensive unconditional reproducibility experiment settings**

| Name | Architecture | SDE | Sampler | Continuous | Distillation |
|------|-------------|-----|---------|------------|--------------|
| DDPMv1 | DDPM | VP | DPM-Solver | ✓ | ✗ |
| DDPMv2 | DDPM | VP | DPM-Solver | ✗ | ✗ |
| DDPMv3 | DDPM++ | VP | DPM-Solver | ✓ | ✗ |
| DDPMv4 | DDPM++(deep) | VP | DPM-Solver | ✓ | ✗ |
| DDPMv5 | DDPM++(deep) | VP | ODE | ✓ | ✗ |
| DDPMv6 | DDPM++(deep) | sub-VP | ODE | ✓ | ✗ |
| DDPMv7 | DDPM | sub-VP | ODE | ✓ | ✗ |
| DDPMv8 | DDPM++ | sub-VP | ODE | ✓ | ✗ |
| Multistagev1 | Multistage (3 stages) | VP | DPM-Solver | ✓ | ✗ |
| Multistagev2 | Multistage (4 stages) | VP | DPM-Solver | ✓ | ✗ |
| Multistagev3 | Multistage (5 stages) | VP | DPM-Solver | ✓ | ✗ |
| EDMv1 | EDM | VP | Heun-Solver | ✓ | ✗ |
| EDMv2 | EDM | VE | Heun-Solver | ✓ | ✗ |
| UViT | UViT | VP | DPM-Solver | ✓ | ✗ |
| DiT | DiT | VP | DPM-Solver | ✓ | ✗ |
| CD | EDM | VE | 1-step | ✓ | ✓ |
| CT | EDM | VE | 1-step | ✓ | ✗ |
| Progressivev1 | DDPM++ | VP | DDIM (1-step) | ✓ | ✓ |
| Progressivev2 | DDPM++ | VP | DDIM (16-step) | ✓ | ✓ |
| Progressivev3 | DDPM++ | VP | DDIM (64-step) | ✓ | ✓ |

tings. To illustrate, we compare DDPMv1 and DDPMv2, demonstrating that model reproducibility remains steadfast across these variations.Moreover, it's worth noting that while all reproducibility scores surpass a threshold of 0.6, signifying robust model reproducibility, some scores do exhibit variations. As highlighted in Figure 14, we observe that similar architectures yield higher reproducibility scores (e.g., DDPMv1-8), models distilled from analogous teacher models exhibit enhanced reproducibility (e.g., Progressivev1-3), and models differing solely in their ODE samplers also display elevated reproducibility scores (e.g., DDPMv4, DDPMv5).We hypothesize that the disparities in reproducibility scores are primarily attributed to biases in parameter estimation. These biases may arise from factors such as differences in architecture, optimization strategies, and other variables affecting model training.

## C   COMPARE GAN & VAE

To further investigate this observation within the realm of diffusion models, we extend our assessment to model similarity in Generative Adversarial Networks (GANs) Goodfellow et al. (2014) and Variational Autoencoders (VAEs) Kingma & Welling (2013). We gauge this similarity through the application of a reproducibility score. In our evaluation of GAN-based methods, we contrast two prominent variants: Wasserstein GAN (wGAN) Arjovsky et al. (2017) and Spectral Normalization GAN (SNGAN) Miyato et al. (2018). We conduct this analysis using the CIFAR-10 dataset. Simultaneously, within the realm of VAE-based approaches, we consider both the standard VAE and the Variational Autoencoding Mutual Information Bottleneck (VAMP) model Tomczak & Welling (2018). Our evaluation focuses on the MNIST dataset introduced by Deng LeCun et al. (1998). It's important to note that each model utilized in this analysis was provided by its respective author, and the reproducibility score calculation follows a similar methodology to that applied in the diffusion model experiments. Of particular significance is the fact that the latent space for VAE-based methods is learned through the encoder, and this encoder architecture varies among different models. In this context, our approach involves sampling initial noise from the latent space of one model and employing it for the generation of another. The similarity matrices, presented in Figure 17, collectively indicate a notable absence of reproducibility in both GAN and VAE methods.

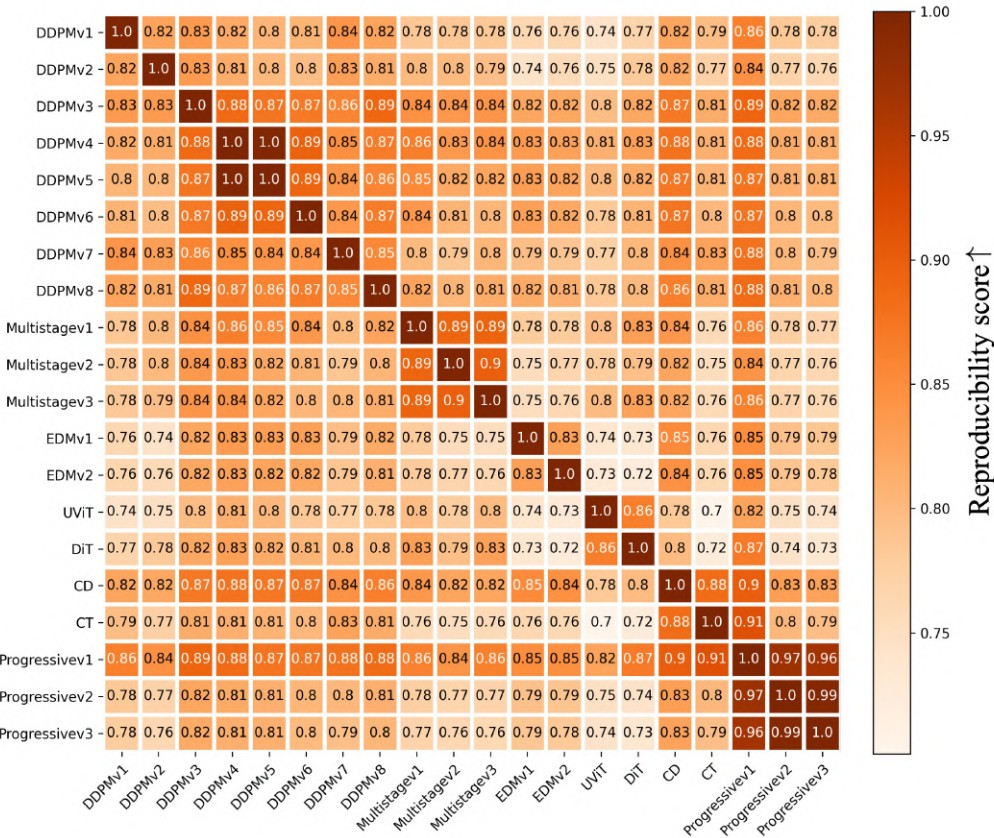

Figure 14: **Comprehensive reproducibility score among different unconditional diffusion model settings.**

## D   THEORETICAL ANALYSIS

This section mainly focus on the proof of Theorem 1 in Section 3. The proof is mainly built upon recent works in Karras et al. (2022).

*Proof.* As the background, let $p_t(\boldsymbol{x}_t|\boldsymbol{x}_0) = \mathcal{N}(\boldsymbol{x}_t; s_t\boldsymbol{x}_0, s_t^2\sigma_t^2\mathbf{I})$ be the perturbation kernel of diffusion model, which is a continuous process gradually adding noise from original image $\boldsymbol{x}_0$ to $\boldsymbol{x}_t$ along the timestep $t \in [0, 1]$. Both $s_t = s(t), \sigma_t = \sigma(t)$ here are simplified as scalar functions of $t$ to control the perturbation kernel. It has been shown that this perturbation kernel is equivalent to a stochastic differential equation $\mathrm{d}\boldsymbol{x} = f(t)\boldsymbol{x}\mathrm{d}t + g(t)\mathrm{d}\boldsymbol{\omega}_t$, where $f(t), g(t)$ are a scalar function of $t$. The relations of $f(t), g(t)$ and $s_t, \sigma_t$ are:

$$s_t = \exp(\int_0^t f(\xi)\mathrm{d}\xi), \text{ and } \sigma_t = \sqrt{\int_0^t \frac{g^2(\xi)}{s^2(\xi)}\mathrm{d}\xi} \qquad (3)$$

Given a dataset $\{\boldsymbol{y}_i\}_{i=1}^N$ with $N$ images, we model the original dataset distribution $p_{\text{data}}$ as multi-Dirac distribution, $p_{\text{data}}(\boldsymbol{x}) = \frac{1}{N}\sum_{i=1}^N \delta(\boldsymbol{x} - \boldsymbol{y}_i)$, the distribution of perturbed image $\boldsymbol{x}$ at random timestep $t$ could be calculated as:

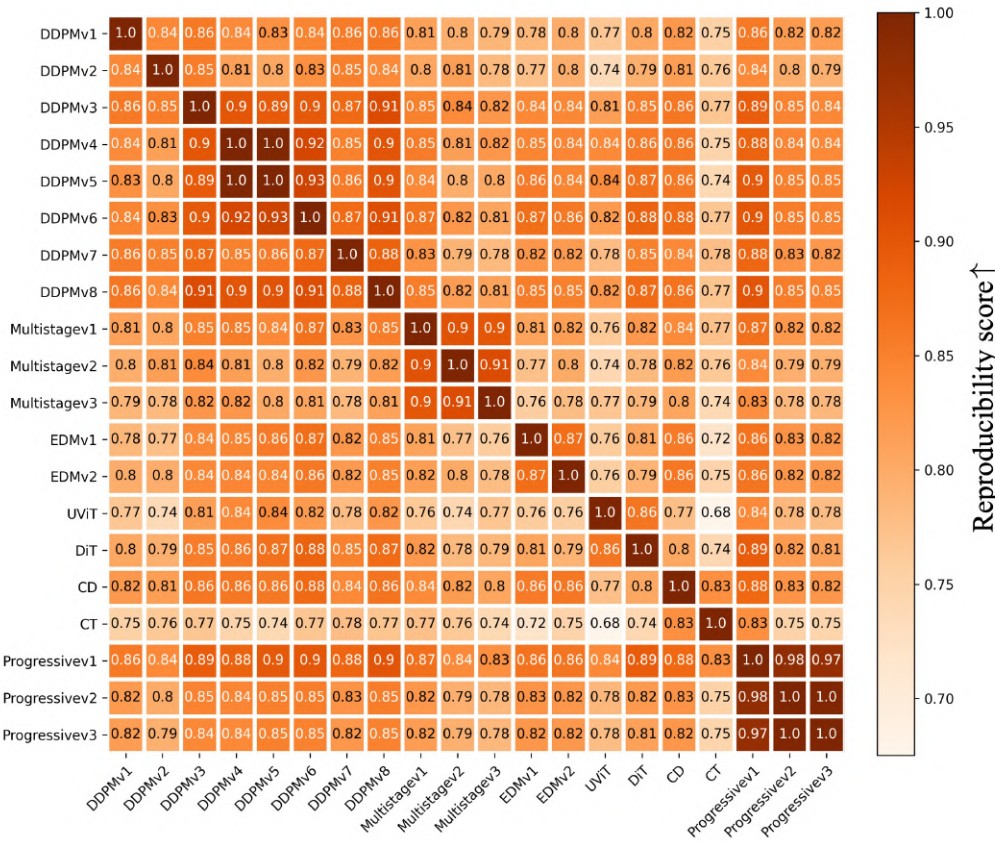

Figure 15: **Comprehensive MAE score among different unconditional diffusion model settings.**

$$p_t(\boldsymbol{x}) = \int_{\mathbb{R}^d} p_t(\boldsymbol{x}|\boldsymbol{x}_0) p_{\text{data}}(\boldsymbol{x}_0) \mathrm{d}\boldsymbol{x}_0 \tag{4}$$

$$= \int_{\mathbb{R}^d} p_{\text{data}}(\boldsymbol{x}_0) \mathcal{N}(\boldsymbol{x}; s_t\boldsymbol{x}_0, s_t^2\sigma_t^2\mathbf{I}) \mathrm{d}\boldsymbol{x}_0 \tag{5}$$

$$= \int_{\mathbb{R}^d} \frac{1}{N} \sum_{i=1}^{N} \delta(\boldsymbol{x}_0 - \boldsymbol{y}_i) \mathcal{N}(\boldsymbol{x}; s_t\boldsymbol{x}_0, s_t^2\sigma_t^2\mathbf{I}) \mathrm{d}\boldsymbol{x}_0 \tag{6}$$

$$= \frac{1}{N} \sum_{i=1}^{N} \int_{\mathbb{R}^d} \delta(\boldsymbol{x}_0 - \boldsymbol{y}_i) \mathcal{N}(\boldsymbol{x}; s_t\boldsymbol{x}_0, s_t^2\sigma_t^2\mathbf{I}) \mathrm{d}\boldsymbol{x}_0 \tag{7}$$

$$= \frac{1}{N} \sum_{i=1}^{N} \mathcal{N}(\boldsymbol{x}; s_t\boldsymbol{y}_i, s_t^2\sigma_t^2\mathbf{I}) \tag{8}$$

Let us consider the noise prediction loss used generally across various diffusion model works:

$$\mathcal{L}(\boldsymbol{\epsilon_\theta}; t) = \mathbb{E}_{\boldsymbol{x} \sim p_t(\boldsymbol{x})}[||\boldsymbol{\epsilon} - \boldsymbol{\epsilon_\theta}(\boldsymbol{x}, t)||^2] \tag{9}$$

$$= \int_{\mathbb{R}_d} \frac{1}{N} \sum_{i=1}^{N} \mathcal{N}(\boldsymbol{x}; s_t\boldsymbol{y}_i, s_t^2\sigma_t^2\mathbf{I}) ||\boldsymbol{\epsilon} - \boldsymbol{\epsilon_\theta}(\boldsymbol{x}, t)||^2 \mathrm{d}\boldsymbol{x} \tag{10}$$

where $\boldsymbol{\epsilon} \sim \mathcal{N}(\mathbf{0}, \mathbf{I})$ is defined follow the perturbation kernel $p_t(\boldsymbol{x}|\boldsymbol{x}_0) = \mathcal{N}(\boldsymbol{x}; s_t\boldsymbol{x}_0, s_t^2\sigma_t^2\mathbf{I})$:

$$\boldsymbol{x} = s_t\boldsymbol{y}_i + s_t\sigma_t\boldsymbol{\epsilon} \Rightarrow \boldsymbol{\epsilon} = \frac{\boldsymbol{x} - s_t\boldsymbol{y}_i}{s_t\sigma_t} \tag{11}$$

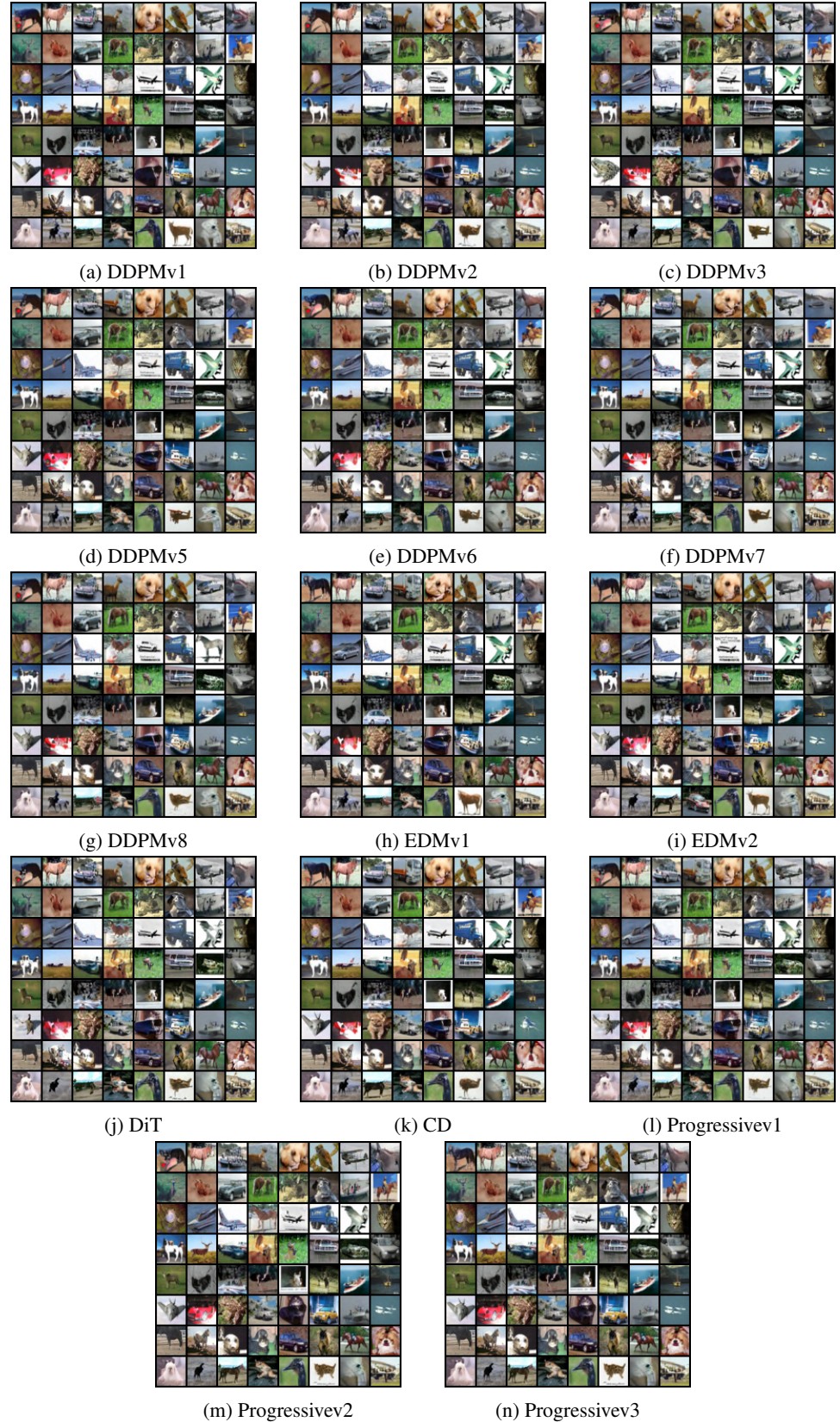

Figure 16: **Comprehensive samples visulization for unconditional diffusion model**

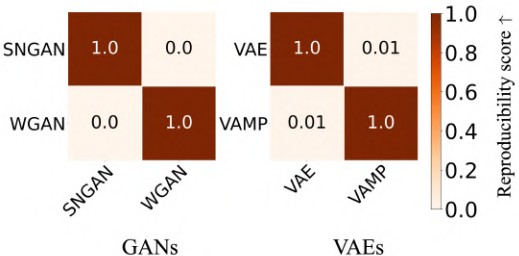

Figure 17: **Quantitative results for GANS and VAEs.**

And $\epsilon_{\boldsymbol{\theta}}$ is a "denoiser" network for learning the noise $\boldsymbol{\epsilon}$.

So plugging Eq. 11 into 10, we could reparameterization the loss as:

$$\mathcal{L}(\epsilon_{\boldsymbol{\theta}};t) = \int_{\mathbb{R}_d} \underbrace{\frac{1}{N}\sum_{i=1}^{N}\mathcal{N}(\boldsymbol{x};s_t\boldsymbol{y}_i,s_t^2\sigma_t^2\mathbf{I})||\epsilon_{\boldsymbol{\theta}}(\boldsymbol{x},t) - \frac{\boldsymbol{x}-s_t\boldsymbol{y}_i}{s_t\sigma_t}||^2}_{=:\mathcal{L}(\epsilon_{\boldsymbol{\theta}};\boldsymbol{x},t)}\,\mathrm{d}\boldsymbol{x} \tag{12}$$

Eq. 12 means we could minimize $\mathcal{L}(\epsilon_{\boldsymbol{\theta}};t)$ by minimizing $\mathcal{L}(\epsilon_{\boldsymbol{\theta}};\boldsymbol{x},t)$ for each $\boldsymbol{x}$. And to find the "optimal denoiser" $\epsilon_{\boldsymbol{\theta}}^*$ that minimize the $\mathcal{L}(\epsilon_{\boldsymbol{\theta}};\boldsymbol{x},t)$ for every given $\boldsymbol{x},t$:

$$\epsilon_{\boldsymbol{\theta}}^*(\boldsymbol{x};t) = \arg\min_{\epsilon_{\boldsymbol{\theta}}(\boldsymbol{x};t)}\mathcal{L}(\epsilon_{\boldsymbol{\theta}};\boldsymbol{x},t) \tag{13}$$

This is a convex optimization problem; the solution could be solved by setting the gradient of $\mathcal{L}(\epsilon_{\boldsymbol{\theta}};\boldsymbol{x},t)$ w.r.t $\epsilon_{\boldsymbol{\theta}}(\boldsymbol{x};t)$ to zero:

$$\nabla_{\epsilon_{\boldsymbol{\theta}}(\boldsymbol{x};t)}[\mathcal{L}(\epsilon_{\boldsymbol{\theta}};\boldsymbol{x},t)] = 0 \tag{14}$$

$$\Rightarrow \nabla_{\epsilon_{\boldsymbol{\theta}}(\boldsymbol{x};t)}[\frac{1}{N}\sum_{i=1}^{N}\mathcal{N}(\boldsymbol{x};s_t\boldsymbol{y}_i,s_t^2\sigma_t^2\mathbf{I})||\epsilon_{\boldsymbol{\theta}}(\boldsymbol{x},t)-\frac{\boldsymbol{x}-s_t\boldsymbol{y}_i}{s_t\sigma_t}||^2] = 0 \tag{15}$$

$$\Rightarrow \frac{1}{N}\sum_{i=1}^{N}\mathcal{N}(\boldsymbol{x};s_t\boldsymbol{y}_i,s_t^2\sigma_t^2\mathbf{I})[\epsilon_{\boldsymbol{\theta}}^*(\boldsymbol{x};t)-\frac{\boldsymbol{x}-s_t\boldsymbol{y}_i}{s_t\sigma_t}] = 0 \tag{16}$$

$$\Rightarrow \epsilon_{\boldsymbol{\theta}}^*(\boldsymbol{x};t) = \frac{1}{s_t\sigma_t}[\boldsymbol{x}-s_t\frac{\sum_{i=1}^{N}\mathcal{N}(\boldsymbol{x};s_t\boldsymbol{y}_i,s_t^2\sigma_t^2\mathbf{I})\boldsymbol{y}_i}{\sum_{i=1}^{N}\mathcal{N}(\boldsymbol{x};s_t\boldsymbol{y}_i,s_t^2\sigma_t^2\mathbf{I})}] \tag{17}$$

It is obvious that the optimal denoiser $\epsilon_{\boldsymbol{\theta}}^*(\boldsymbol{x};t)$ is a function only depend on the perturbation kernel parameter $s_t,\sigma_t$ and the dataset $\{\boldsymbol{y}_i\}_{i=1}^{N}$.

Given the assumption that the denoiser $\epsilon_{\boldsymbol{\theta}}$ could converge to the optimal denoiser $\epsilon_{\boldsymbol{\theta}}^*(\boldsymbol{x};t)$ we use a deterministic ODE sampler to generate images, then mapping $f : \mathcal{E} \mapsto \mathcal{I}$, from gaussion noise space $\mathcal{E}$ to image space $\mathcal{I}$ is an invertible mapping and the inverse mapping $f^{-1}$ is a unique identifiable encoding.

The mapping $f$ is only determined by the $\epsilon_{\boldsymbol{\theta}}^*(\boldsymbol{x};t)$ and the ODE sampler. Take the probability flow ODE sampler Song et al. (2020a) as an example, the ODE is given as:

$$\frac{\mathrm{d}\boldsymbol{x}_t}{\mathrm{d}t} = f(t)\boldsymbol{x}_t + \frac{g^2(t)}{2s_t\sigma_t}\epsilon_{\boldsymbol{\theta}}(\boldsymbol{x}_t;t) \tag{18}$$

The mapping $f$ given the optimal denoiser $\epsilon_{\boldsymbol{\theta}}^*$ could be determined as:

$$f : \mathcal{E} \mapsto \mathcal{I} := \begin{cases} \dfrac{\mathrm{d}\boldsymbol{x}_t}{\mathrm{d}t} = f(t)\boldsymbol{x}_t + \dfrac{g^2(t)}{2s_t\sigma_t}\boldsymbol{\epsilon}_{\boldsymbol{\theta}}^*(\boldsymbol{x}_t; t)\mathrm{d}t \\[2ex] \boldsymbol{\epsilon}_{\boldsymbol{\theta}}^*(\boldsymbol{x}; t) = \dfrac{1}{s_t\sigma_t}[\boldsymbol{x} - s_t \dfrac{\sum_{i=1}^N \mathcal{N}(\boldsymbol{x}; s_t\boldsymbol{y}_i, s_t^2\sigma_t^2\mathbf{I})\boldsymbol{y}_i}{\sum_{i=1}^N \mathcal{N}(\boldsymbol{x}; s_t\boldsymbol{y}_i, s_t^2\sigma_t^2\mathbf{I})}] \\[2ex] \boldsymbol{x}_{t=0} \in \mathcal{I},\ \boldsymbol{x}_{t=1} \in \mathcal{E},\ t : 1 \mapsto 0 \end{cases} \tag{19}$$

The mapping $f$ is invertible and the inverse mapping $f^{-1}$ is defined as:

$$f^{-1} : \mathcal{I} \mapsto \mathcal{E} := \begin{cases} \dfrac{\mathrm{d}\boldsymbol{x}_t}{\mathrm{d}t} = f(t)\boldsymbol{x}_t + \dfrac{g^2(t)}{2s_t\sigma_t}\boldsymbol{\epsilon}_{\boldsymbol{\theta}}^*(\boldsymbol{x}_t; t)\mathrm{d}t \\[2ex] \boldsymbol{\epsilon}_{\boldsymbol{\theta}}^*(\boldsymbol{x}; t) = \dfrac{1}{s_t\sigma_t}[\boldsymbol{x} - s_t \dfrac{\sum_{i=1}^N \mathcal{N}(\boldsymbol{x}; s_t\boldsymbol{y}_i, s_t^2\sigma_t^2\mathbf{I})\boldsymbol{y}_i}{\sum_{i=1}^N \mathcal{N}(\boldsymbol{x}; s_t\boldsymbol{y}_i, s_t^2\sigma_t^2\mathbf{I})}] \\[2ex] \boldsymbol{x}_{t=0} \in \mathcal{I},\ \boldsymbol{x}_{t=1} \in \mathcal{E},\ t : 0 \mapsto 1 \end{cases} \tag{20}$$

For the mapping $f^{-1}$ a clear image $\boldsymbol{x}_{t=0}$ would generate a deterministic embedding noise $\boldsymbol{x}_{t=1}$. If ignoring the discretization error from the ODE sampler, the reverse mapping $f$ could numerically also start from $\boldsymbol{x}_{t=1}$ to generate the specific image $\boldsymbol{x}_{t=0}$ Su et al. (2022). So the encoding is unique. All the above processes could be expressed by the optimal denoiser and ODE solver, which is identifiable. In conclusion, the mapping $f$ of the diffusion model is uniquely identifiable encoding theoretically. □

**Extended Experiment Setting** Similar to what is illustrated in Figure 5, we employ the theoretical generation process, facilitated by the inverse mapping $f^{-1}$. To expedite the sampling speed, we leverage the DPM-Solver Lu et al. (2022).For a more comprehensive view of our results, we present additional visualizations in Figure 18 and Figure 19. In these experiments, we train UNet models with varying numbers of channels on subsets of the CIFAR-10 dataset, each comprising different training samples. Our standard batch size for all experiments is set at 128, and we continue training until the generated samples reach visual convergence, characterized by minimal changes in both appearance and semantic information.

# E  CONDITIONAL DIFFUSION MODEL

**Extended Experiment setting** To investigate the reproducibility of the conditional diffusion model, we opted for three distinct architectures: the conditional EDM Karras et al. (2022), conditional multistage EDM Anonymous, and conditional U-ViT Bao et al. (2023). Our training data consisted of the CIFAR-10 dataset, with the class labels serving as conditions. It's worth noting that the primary distinction between EDM and multistage EDM lies in the architecture of the score function. Conversely, the contrast between EDM and conditional U-ViT extends beyond architectural differences to encompass conditional embeddings. Specifically, EDM transforms class labels into one-hot vectors, subjects them to a single-layer Multilayer Perceptron (MLP), and integrates the output with timestep embeddings. In contrast, U-ViT handles class labels by embedding them through a trainable lookup table, concatenating them with other inputs, including timestep information and noisy image patches represented as tokens. For all three architectures, we pursued training until convergence was achieved, marked by the lowest FID. The DPM-Solver was employed for sampling purposes. To generate samples, we employed the same 10K initial noise distribution as utilized in the unconditional setting (refer to Section 2.1). For each such initial noise instance, we generated 10 images, guided by 10 distinct classes, resulting in a total of 100K images.

**Discussion** The observed reproducibility between the unconditional diffusion model and the conditional diffusion model presents an intriguing phenomenon. It appears that the conditional diffusion model learns a mapping function, denoted as $f_{c\in\mathcal{C}} : \mathcal{E} \mapsto \mathcal{I}_{c\in\mathcal{C}}$, which maps from the same noise space $\mathcal{E}$ to each individual image manifold $\mathcal{I}_{c\in\mathcal{C}}$ corresponding to each class $c$. In contrast, the mapping of the unconditional diffusion model, denoted as $f : \mathcal{E} \mapsto \mathcal{I}$, maps the noise space to a broader image manifold $\mathcal{I} \subset \bigcup_{c\in\mathcal{C}} \mathcal{I}_c$. A theoretical analysis of this unique reproducibility relationship holds the promise of providing valuable insights.

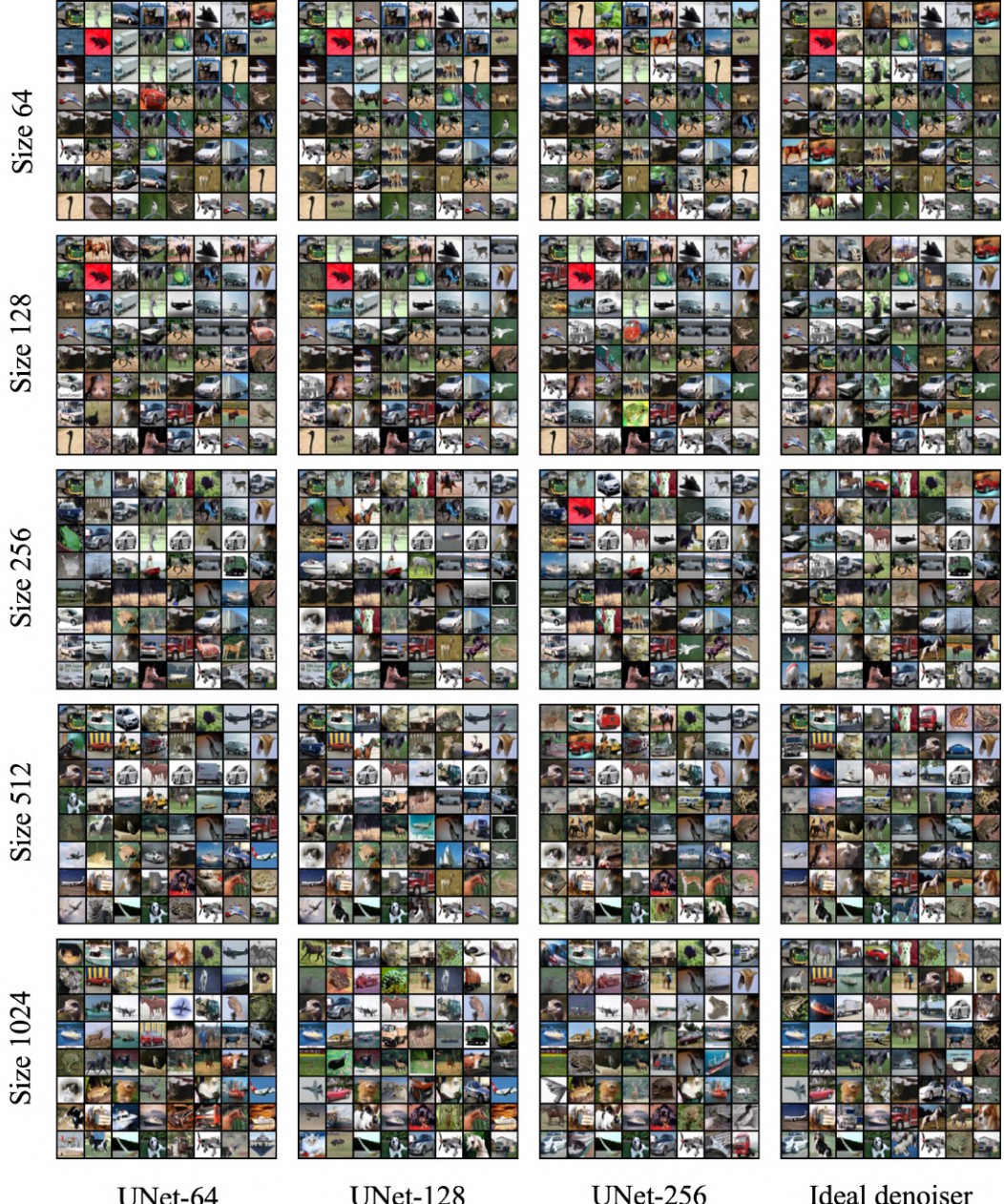

Figure 18: **Visualization between theoretical and experimental results.**

Currently, our research is exclusively focused on the conditional diffusion model. It raises the question of how the reproducibility phenomenon manifests in the context of the text-to-image diffusion model (Rombach et al., 2022a; Ramesh et al., 2021; Nichol et al., 2021), where the conditioning factor is not confined to finite classes but instead involves complex text embeddings.

As illustrated in Figure 20 and Figure 21, our previous comparisons were made with the same initial noise and class conditions. However, when comparing the same model with identical initial noise but different class conditions, we uncovered intriguing findings. For instance, the first row and column images in Figure 20 (i) and (l) exhibited remarkable similarity in low-level structural attributes, such as color, despite differing in semantics. This observation is consistent with findings in Figure 26, where we explored generation using diffusion models trained on mutually exclusive CIFAR-100 and CIFAR-10 datasets. These findings bear a striking resemblance to the conclusions drawn in Khrulkov et al. (2022), which also demonstrated a similar phenomenon in a simplified scenario,

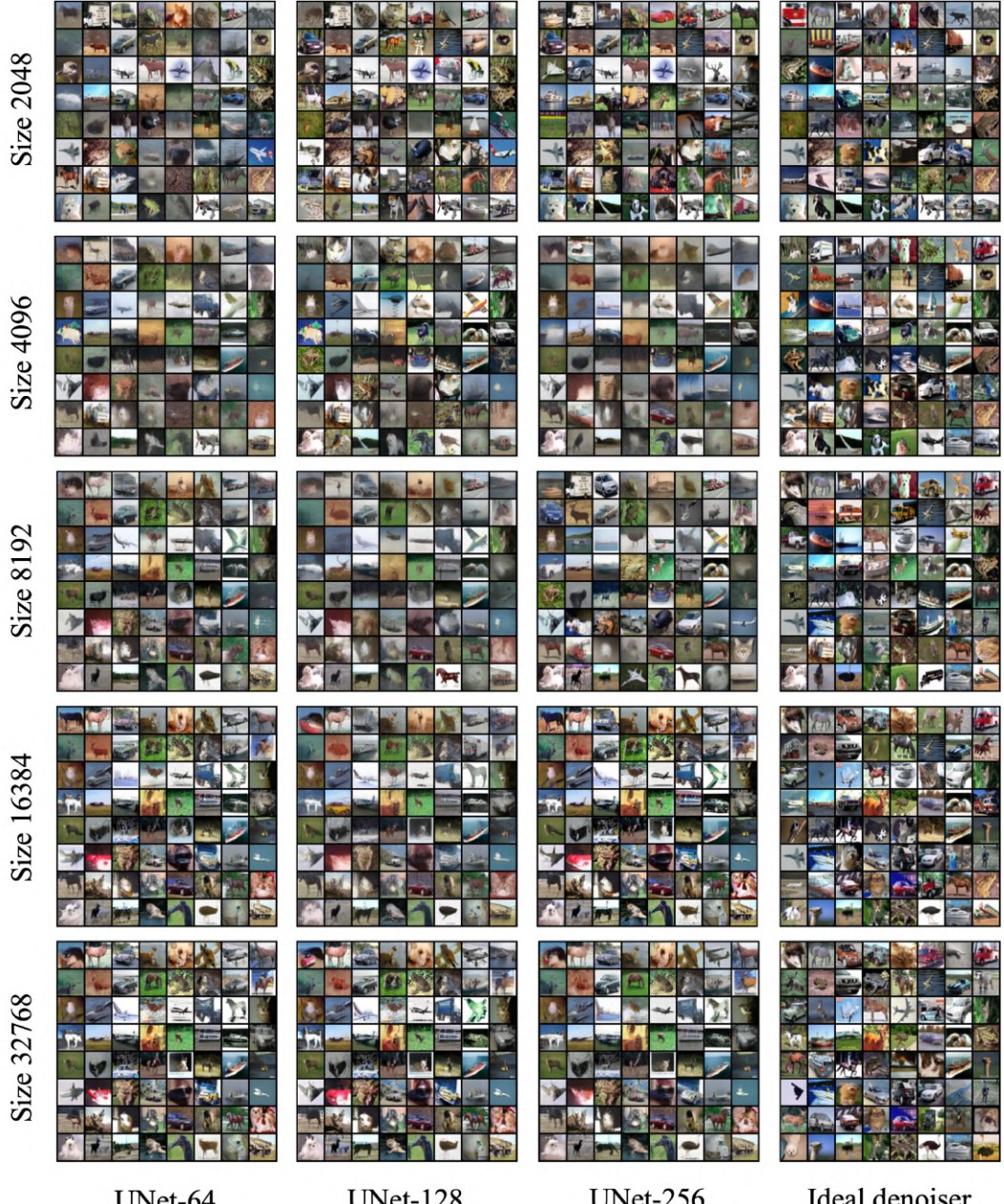

Figure 19: **Visualization between theoretical and experimental results.**

where $\mathcal{I}$ follows a Gaussian distribution. To gain a deeper understanding of reproducibility and the phenomena mentioned in this paragraph, leveraging optimal transport methods (e.g., Schrödinger bridge (Shi et al., 2023; De Bortoli et al., 2021; Luo et al., 2023b; Delbracio & Milanfar, 2023; Liu et al., 2023)) holds significant potential.

## F    DIFFUSION MODEL FOR SOLVING INVERSE PROBLEM

To explore the reproducibility of diffusion models in solving inverse problems, we adopted the Diffusion Posterior Sampling (DPS) strategy proposed by Chung et al. Chung et al. (2022a). Our adaptation involved a slight modification of their algorithm, specifically by eliminating all sources of stochasticity within it. Additionally, we employed the DPM-Solver for Diffusion Posterior Sampling.

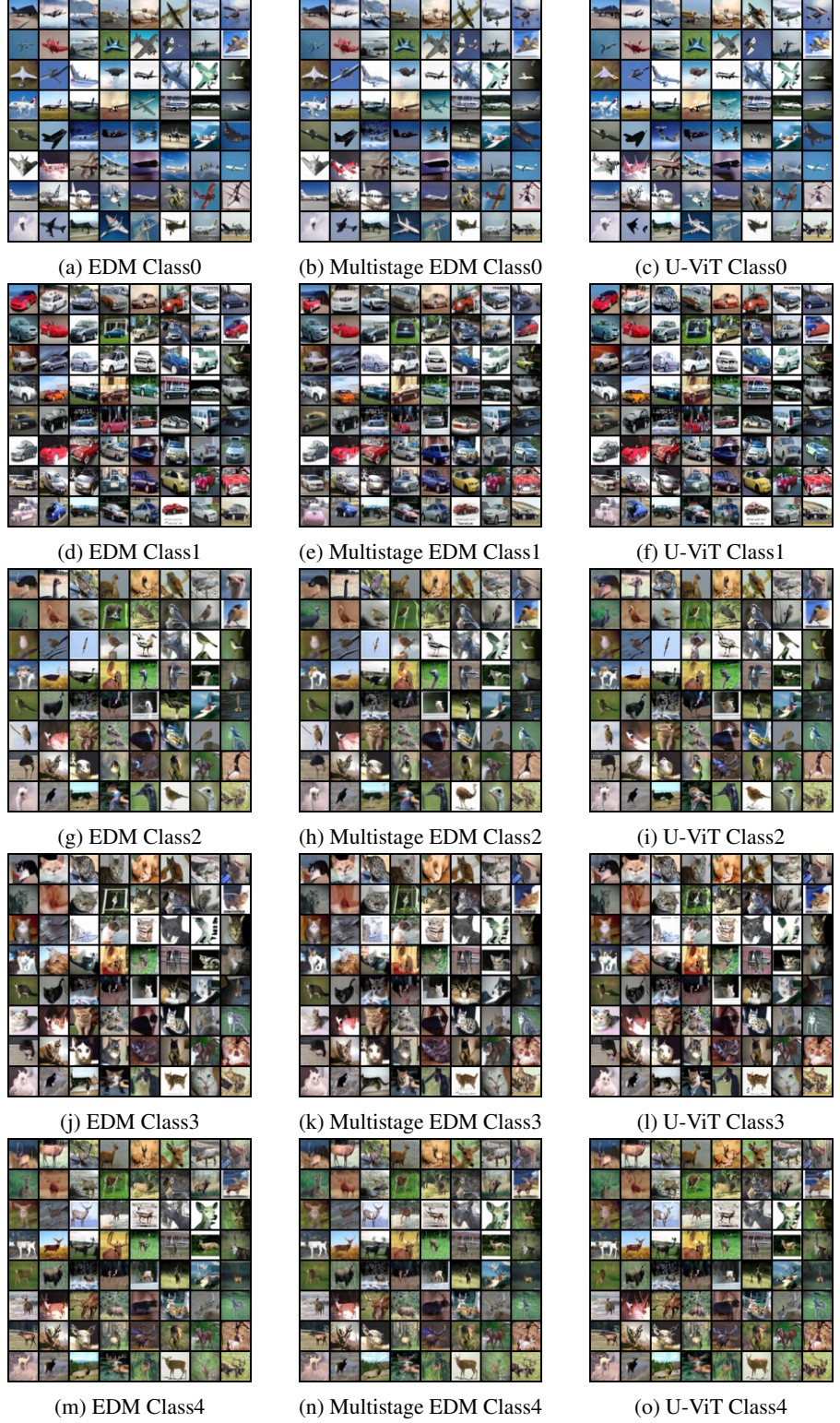

(a) EDM Class0     (b) Multistage EDM Class0     (c) U-ViT Class0

(d) EDM Class1     (e) Multistage EDM Class1     (f) U-ViT Class1

(g) EDM Class2     (h) Multistage EDM Class2     (i) U-ViT Class2

(j) EDM Class3     (k) Multistage EDM Class3     (l) U-ViT Class3

(m) EDM Class4     (n) Multistage EDM Class4     (o) U-ViT Class4

Figure 20: **Visualization of conditional diffusion model generations (class 0 - 4).**

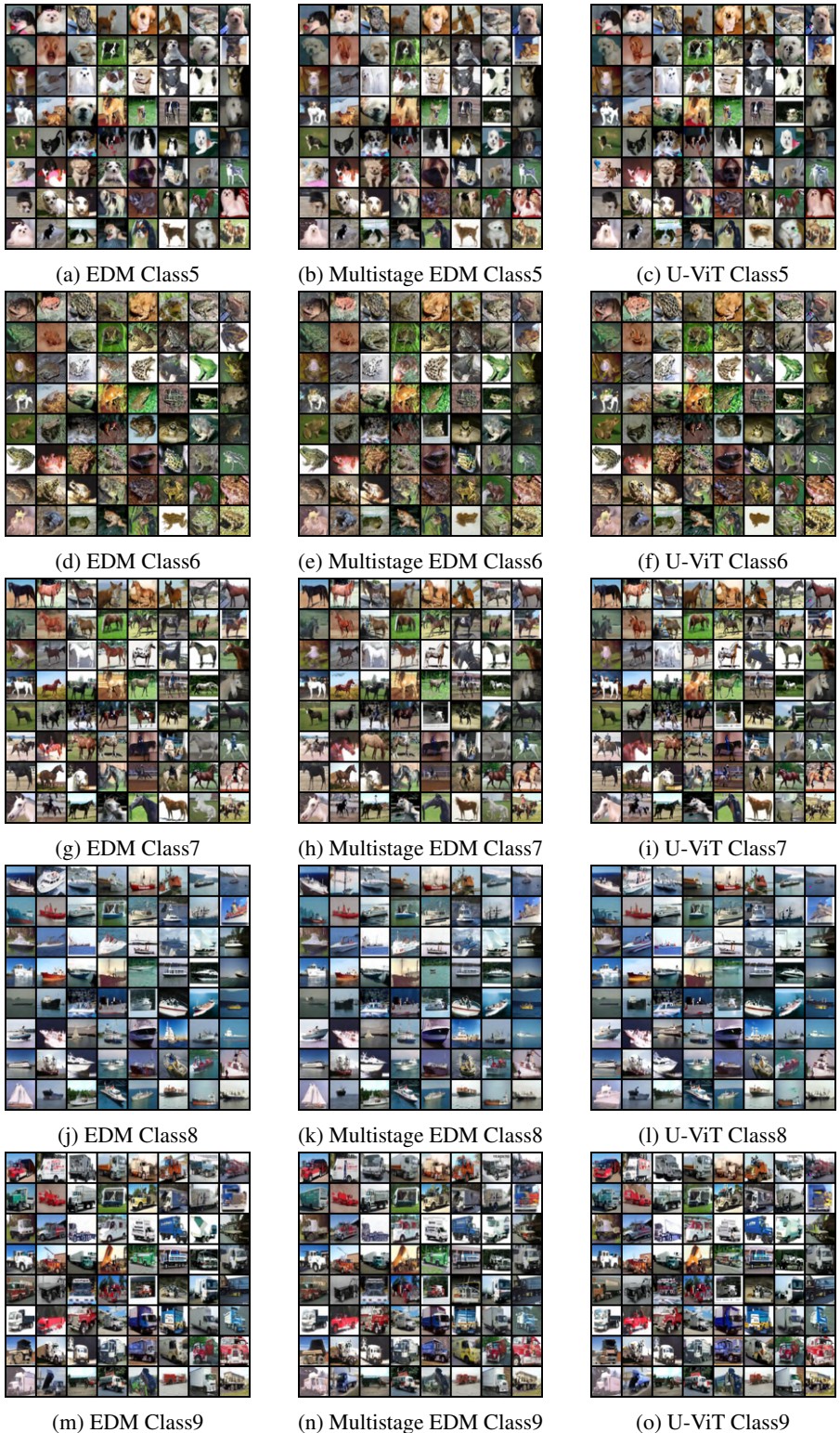

(a) EDM Class5     (b) Multistage EDM Class5     (c) U-ViT Class5

(d) EDM Class6     (e) Multistage EDM Class6     (f) U-ViT Class6

(g) EDM Class7     (h) Multistage EDM Class7     (i) U-ViT Class7

(j) EDM Class8     (k) Multistage EDM Class8     (l) U-ViT Class8

(m) EDM Class9     (n) Multistage EDM Class9     (o) U-ViT Class9

Figure 21: **Visualization of conditional diffusion model generations (class 5 - 9).**

**Extended Experiment setting** To explore the reproducibility of diffusion models in solving inverse problems, we adopted the Diffusion Posterior Sampling (DPS) strategy proposed by Chung et al. Chung et al. (2022a). Our adaptation involved a slight modification of their algorithm, specifically by eliminating all sources of stochasticity within it. Additionally, we employed the DPM-Solver for

Diffusion Posterior Sampling: Algorithm 1, with $N_{\text{dps}} = 34$ posterior samping steps, 33 iterations for 3rd order DPM-Solver, 1 for 1st order DPM-Solver, thus 100 function evaluations. We also set all $\xi_i = 1$.

For the task involving image inpainting on the CIFAR-10 dataset, we applied two square masks to the center of the images. One mask measured 16 by 16 pixels, covering 25% of the image area, and the other measured 25 by 25 pixels, covering 61% of the image area. We denoted these as "easy inpainting" and "hard inpainting" tasks. In Figure 7 and Figure 22, we utilized the "easy inpainting" scenario with a specific observation $z$ as illustrated in the figure. In Figure 25, we considered both the "easy inpainting" and "hard inpainting" tasks. We also employed 10K distinct initial noise and their corresponding 10K distinct observations $z$ to calculate the reproducibility score, as presented in Figure 25. Additional visualizations for Figure 7 and Figure 22 are provided in Figure 23 and Figure 24.

---

**Algorithm 1** Determinsitic DPS with DPM-Solver.

---

**Require:** $N_{\text{dps}}$, $\boldsymbol{u}$, $f(t)$, $g(t)$, $s_t$, $\sigma_t$, $\{\xi_i\}_{i=1}^{N_{\text{dps}}}$
1: $\boldsymbol{x}_{N_{\text{dps}}} \sim \mathcal{N}(\mathbf{0}, \boldsymbol{I})$
2: **for** $i = N_{\text{dps}}$ **to** $q$ **do**
3: $\quad \hat{\boldsymbol{x}}_0 = \dfrac{1}{f(i)}\left(\boldsymbol{x}_i - \dfrac{g^2(i)}{s_i \sigma_i}\boldsymbol{\epsilon_\theta}\left(\boldsymbol{x}_i, i\right)\right)$
4: $\quad \boldsymbol{x}'_{i-1} \leftarrow \text{Dpm-Solver}(\boldsymbol{x}_i, i)$
5: $\quad \boldsymbol{x}_{i-1} \leftarrow \boldsymbol{x}'_{i-1} - \xi_i \nabla_{\boldsymbol{x}_i}||\boldsymbol{u} - \mathcal{A}\left(\hat{\boldsymbol{x}}_0\right)||_2^2$
6: **end for**
7: **return** $\hat{\mathrm{x}}_0$

---

**Discussion** Reproducibility is a highly desirable property when employing diffusion models to address inverse problems, particularly in contexts such as medical imaging where it ensures the reliability of generated results. As observed in Figure 22, the reproducibility scores vary for different observations $z$, and the decrease in reproducibility differs across various architecture categories. For instance, when considering observation $z_1$, the reproducibility scores across different architecture categories remain above 0.5, whereas for $z_3$, they fall below 0.3. Since the choice of observation $z$ also significantly impacts reproducibility, we conducted a complementary experiment presented in Figure 25. In this experiment, for each initial noise instance, we employed a different observation $z$. From the results, it is evident that reproducibility decreases between different categories of diffusion models. Furthermore, reproducibility diminishes as the inpainting task becomes more challenging, with "hard inpainting" being more demanding than "easy inpainting."

Here is an intuitive hypothesis of the decreasing reproducibility:

The update step of Diffusion Posterior Sampling (DPS), is constrained by the data consistency through the following equation:

$$\boldsymbol{x}_{i-1} \leftarrow \boldsymbol{x}'_{i-1} - \xi_i \nabla_{\boldsymbol{x}_i}||\boldsymbol{u} - \mathcal{A}\left(\hat{\boldsymbol{x}}_0\right)||_2^2 \tag{21}$$

Where $\hat{\boldsymbol{x}}_0 = \dfrac{1}{f(i)}\left(\boldsymbol{x}_i - \dfrac{g^2(i)}{s_i \sigma_i}\boldsymbol{\epsilon_\theta}\left(\boldsymbol{x}_i, i\right)\right)$, we could show that:

$$\xi_i \nabla_{\boldsymbol{x}_i}||\boldsymbol{z} - \mathcal{A}\left(\hat{\boldsymbol{x}}_0\right)||_2^2 = \frac{\partial \mathcal{A}\left(\hat{\boldsymbol{x}}_0\right)}{\partial \boldsymbol{x}_i}\left(\mathcal{A}\left(\hat{\boldsymbol{x}}_0\right) - \boldsymbol{z}\right) \tag{22}$$

$$= \frac{\partial \mathcal{A}\left(\hat{\boldsymbol{x}}_0\right)}{\partial \hat{\boldsymbol{x}}_0}\frac{\partial \hat{\boldsymbol{x}}_0}{\partial \boldsymbol{x}_i}\left(\mathcal{A}\left(\hat{\boldsymbol{x}}_0\right) - \boldsymbol{z}\right) \tag{23}$$

$$= \frac{1}{f(i)}\frac{\partial \mathcal{A}\left(\hat{\boldsymbol{x}}_0\right)}{\partial \hat{\boldsymbol{x}}_0}\left(1 - \frac{g^2(i)}{s_i \sigma_i}\frac{\partial \boldsymbol{\epsilon_\theta}\left(\boldsymbol{x}_i, i\right)}{\partial x_i}\right)\left(\mathcal{A}\left(\hat{\boldsymbol{x}}_0\right) - \boldsymbol{z}\right) \tag{24}$$

This analysis highlights that the unconditional diffusion model is reproducible as long as the function $\boldsymbol{\epsilon_\theta}$ is reproducible. However, for the diffusion model used in inverse problems to be reproducible,

both the function $\epsilon_{\boldsymbol{\theta}}\left(\boldsymbol{x}_t, t\right)$ and its first-order derivative with respect to $\boldsymbol{x}_t$ must be reproducible. In other words, the denoiser should exhibit reproducibility not only in its results but also in its gradients. Combining the findings in Figure 25, we can infer that for similar architectures, reproducibility also extends to the gradient space $\dfrac{\partial \epsilon_{\boldsymbol{\theta}}\left(\boldsymbol{x}_t, t\right)}{\partial \boldsymbol{x}_t}$, which may not hold true for dissimilar architectures. Ensuring reproducibility in the gradient space should thus be a significant focus for achieving reproducibility in diffusion models for solving inverse problems.

Additionally, it's worth noting that the data $\boldsymbol{x}_t$ passed into the denoiser $\epsilon_{\boldsymbol{\theta}}\left(\boldsymbol{x}_t, t\right)$ is always out-of-distribution (OOD) data, especially in tasks like image inpainting. Consequently, the reproducibility of OOD data $\boldsymbol{x}_t$ is also crucial for achieving reproducibility in diffusion models for solving inverse problems.

## G   FINE-TUNING DIFFUSION MODEL

**Extended Experiment setting** In our investigation of reproducibility during fine-tuning, we first trained an unconditional diffusion model using EDM Karras et al. (2022) on the CIFAR-100 dataset Krizhevsky et al. (2009). All the fine-tuned models discussed in this section were pre-trained on this model. Subsequently, we examined the impact of dataset size by conducting fine-tuning on the EDM using varying numbers of CIFAR-10 images: 64, 1024, 4096, 16384, and 50000, respectively. Building upon the findings in Moon et al. (2022), which indicate that fine-tuning the attention blocks is less susceptible to overfitting, we opted to target all attention layers for fine-tuning in our experiments. For comparison purposes, we also trained a diffusion model from scratch on the CIFAR-10 dataset, using the same subset of images. All models were trained for the same number of training iterations and were ensured to reach convergence, as evidenced by achieving a low Fréchet Inception Distance (FID) and maintaining consistent mappings from generated samples. The training utilized a batch size of 128 and did not involve any data augmentation.

**Extended Results** Additional generations produced by both the "from scratch" diffusion models and the fine-tuned diffusion models are presented in Figure 26, encompassing various training dataset sizes. A notable observation arises when comparing the fine-tuned diffusion model's generation using 4096 and 50000 data samples. Even with this limited dataset, the fine-tuned diffusion model demonstrates a remarkable ability to approximate the target distribution. This suggests that the fixed portion of the diffusion model, containing information from the pre-trained CIFAR-100 dataset, aids the model in converging to the target distribution with less training data. In contrast, when attempting to train the diffusion model from scratch on CIFAR-10, even with 16384 data samples, it fails to converge to the target distribution. Additionally, despite the distinct nature of CIFAR-100 and CIFAR-10, their generations from the same initial noise exhibit striking similarities (Figure 26). This similarity might be a contributing factor explaining how the pre-trained CIFAR-100 diffusion model assists in fine-tuning the diffusion model to converge onto the CIFAR-10 manifold with reduced training data.

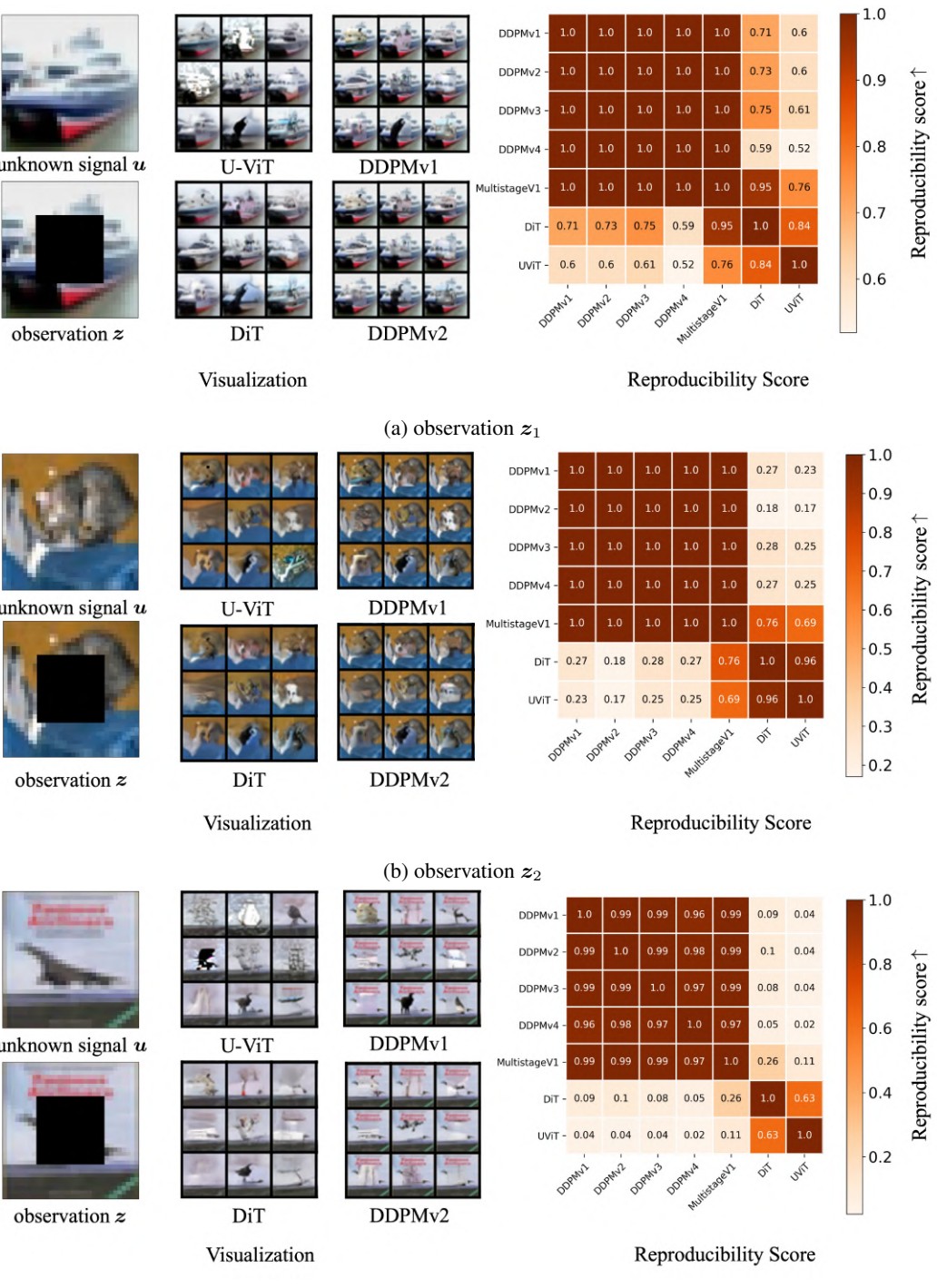

(a) observation $z_1$

(b) observation $z_2$

(c) observation $z_3$

Figure 22: **Visualization of inverse problem solving with different observations**

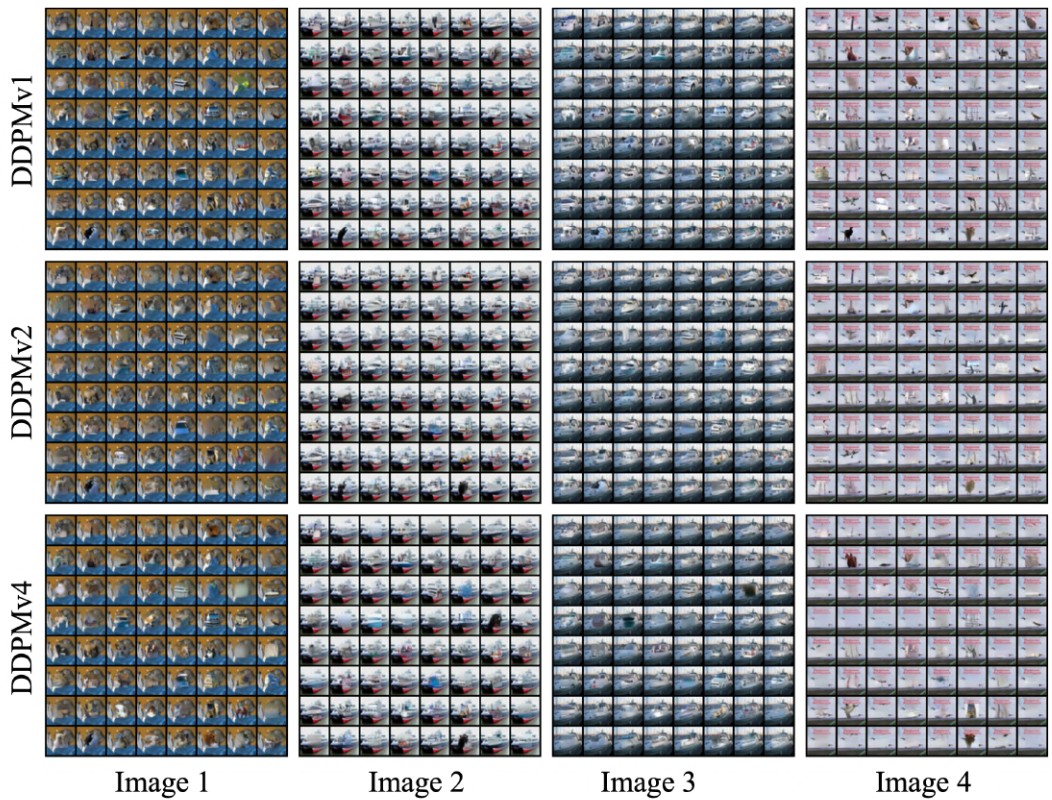

Image 1      Image 2      Image 3      Image 4

Figure 23: **More visualization results for Figure 7 and Figure 22**

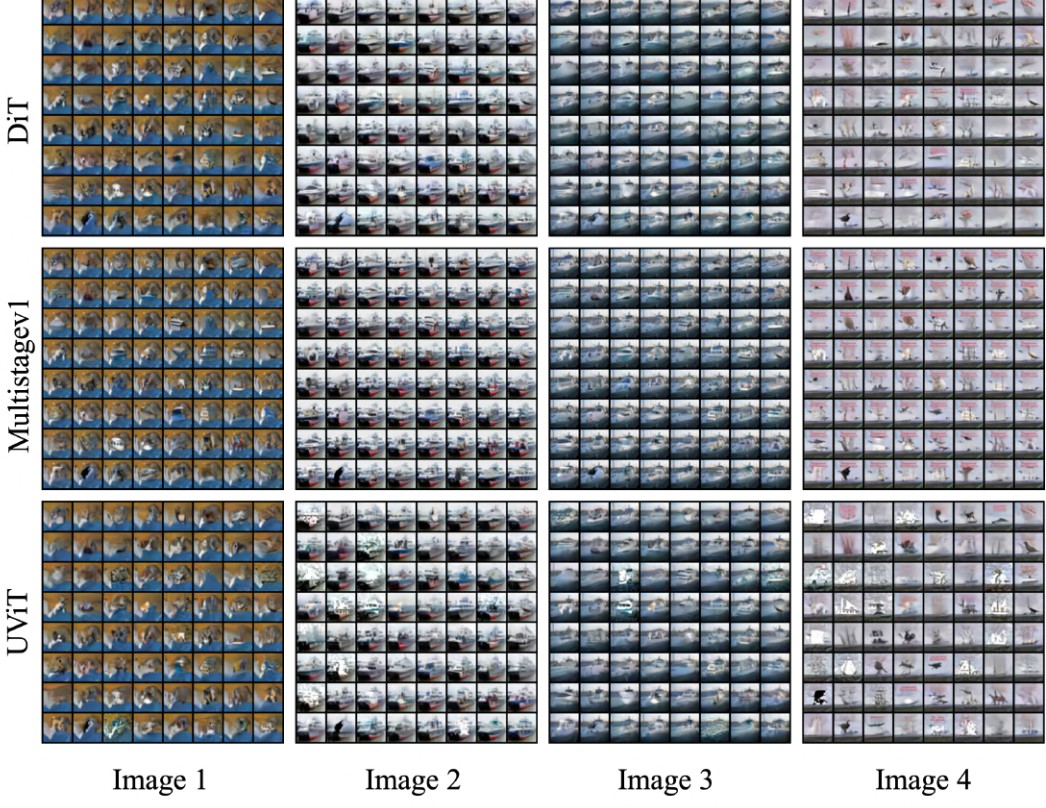

Image 1      Image 2      Image 3      Image 4

Figure 24: **More visualization results for Figure 7 and Figure 22**

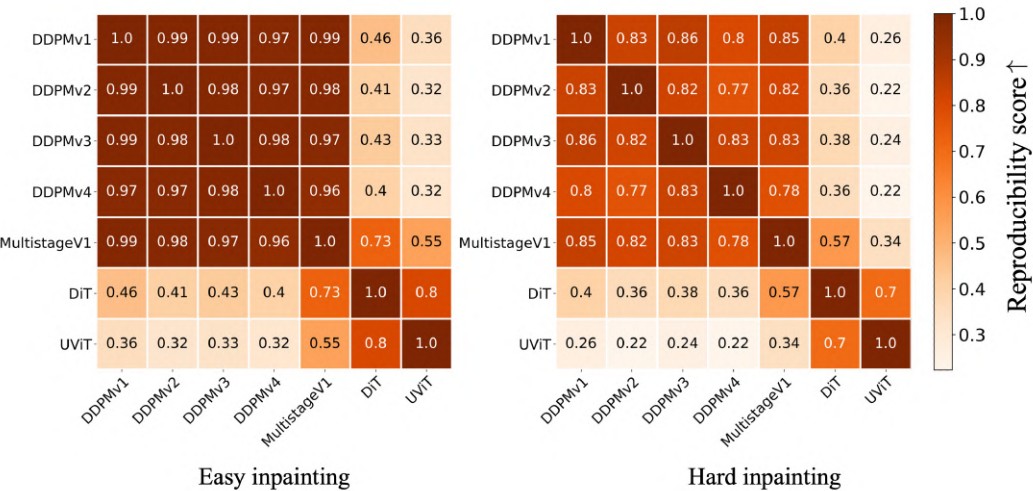

Figure 25: **Extended experiments on image impainting for reproducibility score.**

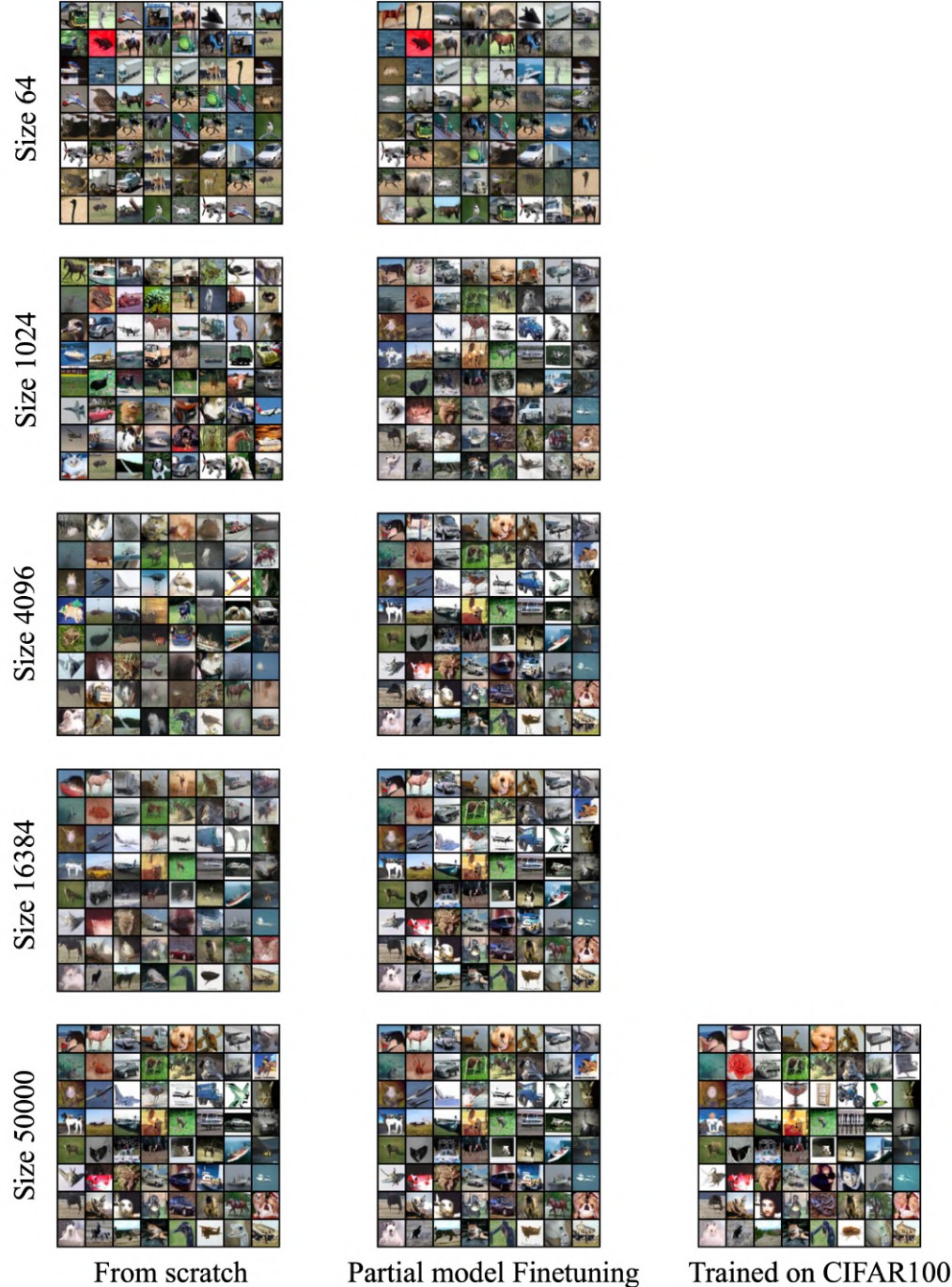

Figure 26: **More visualization of finetuning diffusion models**

