# OpenReview forum: "The Emergence of Reproducibility and Consistency in Diffusion Models"
_ICLR.cc/2024/Conference — Submitted to ICLR 2024_

### Official Review · Reviewer_uEsZ · 2023-10-30

**Soundness:** 2 fair
**Presentation:** 3 good
**Contribution:** 2 fair
**Rating:** 5
**Confidence:** 4

**Summary:**

This paper found that diffusion models for image and text-to-image generation uniquely produce almost identical results when given the same starting conditions, regardless of their design or training. This happens in two ways: either by memorizing training data or by learning broadly from large datasets. This consistent output trait, present in various diffusion model types, could make these models more understandable and controllable.

**Strengths:**

It is quite interesting to investigate the memorization and generalization of diffusion models, which help us to understand current generative model better.

**Weaknesses:**

Although the author discovered an interesting phenomenon and used rich theories and experiments as supplementary explanations. But it seems that there is no further improvement in the training of the current diffusion model.

The experiments are conducted on small datasets (CIFAR)? It is not clear the performance on larger datasets.

**Questions:**

How does the ODE solver and the NFE influence the image? (Few NFE will brings corrupted image)

Have the authors investigated larger datasets, such as CoCo, ImageNet?

How can we get a better training strategy from the current conclusion?

---

> ### Author Response · Authors · 2023-11-15
> **Response to Reviewer uEsZ**
>
> We thank the reviewer for acknowledging the quality of our presentation, experiments, and the timeliness of our results. We value the constructive feedback that the reviewer provided. In the following, we carefully respond to each of the concerns and suggestions that the reviewer raised.
>
>
> ___
>
> > **Q1**:
> > “But it seems that there is no further improvement in the training of the current diffusion model.”
>
> **A1**:
>
> We agree with the reviewer that improving the training of diffusion models is not the main focus of this work. However, we believe that research progress can be driven by two distinct avenues: (i) technical enhancements and (ii) the pursuit of deeper insights. Both aspects hold equal significance. In our investigation, we directed our efforts towards gaining a deeper understanding of diffusion models, unveiling several intriguing phenomena, which we would like to highlight as follows.
>
> -   The first systematical and quantitative study of the model reproducibility phenomenon on the unconditional diffusion model.
>
> -   Moreover, we discovered that model reproducibility occurs in two distinct regimes, the “memorization regime” and “generalizability regime”, and model reproducibility has a strong correlation with model generalizability. Moreover, we have theoretically and experimentally justified the reproducibility in the memorization regime.
>
> -   We generalized the study of model reproducibility in a variety of settings, including conditional diffusion models, diffusion models for inverse problems, and fine-tuning diffusion models. For these different settings, we showed that the model reproducibility manifests in different but principled ways.
>
> Our findings could have a variety of potential applications that we discuss in Q2.
>
> ___
>
> > **Q2**:
> > “How can we get a better training strategy from the current conclusion?”
>
> **A2**:
>
> We thank the reviewer for this important question, and we believe that our understanding opens many interesting application ideas beyond improving training efficiency. One major insight gained from this study is that the mapping from the noise space to the image space is a unique one-to-one mapping, regardless of network architectures or training procedures.
>
>   As we discussed in Section 1, this observation implies that the noise space contains valuable information that can be utilized for controlled and efficient data-generative processes using diffusion models. For instance,
>
>
>
> -  In text-driven image generation, insights into this question could help to guide the content generation (e.g., adversarial attacking [1], robust defending [2], copyright protection [3]) using the same prompt but varying noise inputs.
>
>
> -   Moreover, in the context of solving inverse problems, an answer to this question could guide us to select the input noise for improving the sampling efficiency, and reducing the uncertainty and variance in our signal reconstruction [4, 5, 6].
>
> -   Theoretically, understanding the question will shed light on how the mapping function is learned and constructed between the noise and data distributions, which is crucial for understanding the generation process through the distribution transformation or identifiable encoding [7, 8].
>
>
>
>
> [1] Zou, Andy, Zifan Wang, J. Zico Kolter, and Matt Fredrikson. "Universal and transferable adversarial attacks on aligned language models." arXiv preprint arXiv:2307.15043 (2023).
>
> [2] Zhu, Kaijie, Jindong Wang, Jiaheng Zhou, Zichen Wang, Hao Chen, Yidong Wang, Linyi Yang et al. "PromptBench: Towards Evaluating the Robustness of Large Language Models on Adversarial Prompts." arXiv preprint arXiv:2306.04528 (2023).
>
> [3] Somepalli, Gowthami, Vasu Singla, Micah Goldblum, Jonas Geiping, and Tom Goldstein. "Understanding and Mitigating Copying in Diffusion Models." arXiv preprint arXiv:2305.20086 (2023).
>
> [4] Jalal, Ajil, Marius Arvinte, Giannis Daras, Eric Price, Alexandros G. Dimakis, and Jon Tamir. "Robust compressed sensing mri with deep generative priors." Advances in Neural Information Processing Systems 34 (2021): 14938-14954.
>
> [5] Chung, Hyungjin, and Jong Chul Ye. "Score-based diffusion models for accelerated MRI." Medical image analysis 80 (2022): 102479.
>
> [6] Luo, Guanxiong, Moritz Blumenthal, Martin Heide, and Martin Uecker. "Bayesian MRI reconstruction with joint uncertainty estimation using diffusion models." Magnetic Resonance in Medicine 90, no. 1 (2023): 295-311.
>
> [7] Roeder, Geoffrey, Luke Metz, and Durk Kingma. "On linear identifiability of learned representations." In International Conference on Machine Learning, pp. 9030-9039. PMLR, 2021.
>
> [8] Khemakhem, Ilyes, Diederik Kingma, Ricardo Monti, and Aapo Hyvarinen. "Variational autoencoders and nonlinear ica: A unifying framework." In International Conference on Artificial Intelligence and Statistics, pp. 2207-2217. PMLR, 2020.

---

> ### Author Response · Authors · 2023-11-15
> **Response to Reviewer uEsZ**
>
> > **Q3**: “How does the ODE solver and the NFE influence the image?”
>
> **A3**:
>
> We thank the reviewer for the question. We want to clarify that the model reproducibility is regarding the content or shape of the image, not image quality. Starting from the same noise, we find that different diffusion models generate the same content, but the image quality can be different depending on the model.
>
> To the reviewer’s question, for the ODE solver, we have covered DPM-Solver [1], Heun-Solver [2], and DDIM [3] for unconditional reproducibility. We find that different ODE solvers do not affect model reproducibility, although they may affect the image quality. The experiment settings are in Table 1, Appendix B. The reproducibility scores are shown in Figure 14.
>
> For NFE, we added more experiments in Appendix A.4 during the revision. While a lower NFE tends to degrade the generated image quality, our observations reveal that the content of the generated images remains remarkably consistent across varying NFE levels; see Figure 12.
>
> [1] Lu, Cheng, Yuhao Zhou, Fan Bao, Jianfei Chen, Chongxuan Li, and Jun Zhu. "Dpm-solver: A fast ode solver for diffusion probabilistic model sampling in around 10 steps." _Advances in Neural Information Processing Systems_ 35 (2022): 5775-5787.
>
> [2] Karras, Tero, Miika Aittala, Timo Aila, and Samuli Laine. "Elucidating the design space of diffusion-based generative models." _Advances in Neural Information Processing Systems_ 35 (2022): 26565-26577.
>
> [3] Song, Jiaming, Chenlin Meng, and Stefano Ermon. "Denoising diffusion implicit models." _arXiv preprint arXiv:2010.02502_ (2020).

---

> ### Author Response · Authors · 2023-11-15
> **Response to Reviewer uEsZ**
>
> >**Q4**: “Have the authors investigated larger datasets, such as CoCo, ImageNet?”
>
> **A4**:
> We thank the reviewer for the valuable suggestions to study the phenomenon more comprehensively. During the revision, we conducted more experiments considering two extra cases: (i) the Conditional Diffusion Model on the ImageNet Dataset [1], and (ii) Stable Diffusion [2] on the LAION-5B dataset [3]. The results can be found in Appendix A.1 and A.2 of the revised paper. As such, our new experiments not only covered larger datasets in (i, ii), but also text-to-image diffusion models in (ii). Specifically, we find the following:
>
> 1. Conditional Diffusion Model on the ImageNet Dataset: We evaluated the reproducibility of EDM [4] and ADM [5], as shown in the following table:
>
> 	| $\text{RP} \text{ Score}$| EDM  | ADM  |
> 	|----------|------|------|
> 	| **EDM**      | 1.0  | 0.81 |
> 	| **ADM**      | 0.81 | 1.0  |
>
> 	Detailed experimental settings and visual results are presented in Appendix A.1. We find that the model reproducibility still exists on ImageNet.
>
> 2. Stable Diffusion on the LAION-5B Dataset: We evaluated the reproducibility of stable diffusion models with versions v1-1 to v1-4, with their respective reproducibility scores shown in the table below:
>
> 	| $\text{RP} \text{ Score}$ | V1-1 | V1-2 | V1-3 | V1-4 |
> 	|----------|------|------|------|------|
> 	| **V1-1**     | 1.0  | 0.10 | 0.03 | 0.03 |
> 	| **V1-2**     | 0.10 | 1.0  | 0.21 | 0.20 |
> 	| **V1-3**     | 0.03 | 0.21 | 1.0  | **0.63** |
> 	| **V1-4**     | 0.03 | 0.20 | **0.63** | 1.0  |
>
> 	The relationship between V1-1 to V1-4 could be summarized as
> 	- Versions v1-1, v1-2, and v1-3 each are trained on different subsets of the LAION-5B dataset.
> 	- Versions v1-3 and v1-4 share the same training subset from LAION-5B.
> 	- Version v1-2 is resumed from v1-1, while v1-3 and v1-4 are resumed from v1-2.
>
> 	Notably, only versions v1-3 and v1-4 were trained on the exact same dataset, resulting in the highest reproducibility scores (0.63). Lesser but noticeable reproducibility scores (below 0.21) are observed among v1-1, v1-2, and v1-3, this can be attributed to their sequential training and overlapping datasets. There is no model reproducibility between v1-1 and others, because the training datasets are nonoverlapping and different. Further details on the experimental setup and visual results can be found in Appendix A.2.
>
> In conclusion, our findings suggest that this phenomenon persists for larger diffusion models trained on large datasets.
>
> [1]Deng, Jia, Wei Dong, Richard Socher, Li-Jia Li, Kai Li, and Li Fei-Fei. "Imagenet: A large-scale hierarchical image database." In 2009 IEEE conference on computer vision and pattern recognition, pp. 248-255. Ieee, 2009.
>
> [2]Rombach, Robin, Andreas Blattmann, Dominik Lorenz, Patrick Esser, and Björn Ommer. "High-resolution image synthesis with latent diffusion models." In Proceedings of the IEEE/CVF conference on computer vision and pattern recognition, pp. 10684-10695. 2022.
>
> [3] Schuhmann, Christoph, Romain Beaumont, Richard Vencu, Cade Gordon, Ross Wightman, Mehdi Cherti, Theo Coombes et al. "Laion-5b: An open large-scale dataset for training next generation image-text models." Advances in Neural Information Processing Systems 35 (2022): 25278-25294.
>
> [4] Karras, Tero, Miika Aittala, Timo Aila, and Samuli Laine. "Elucidating the design space of diffusion-based generative models." _Advances in Neural Information Processing Systems_ 35 (2022): 26565-26577.
>
> [5] Dhariwal, Prafulla, and Alexander Nichol. "Diffusion models beat gans on image synthesis." _Advances in neural information processing systems_ 34 (2021): 8780-8794.

---

> ### Author Response · Authors · 2023-11-18
> **Looking forward to your feedback**
>
> Dear Reviewer uEsZ:
>
> I would like to begin by expressing my sincere gratitude for the time and effort you have dedicated to reviewing our work. Your insights and feedback are invaluable to us. We hope our responses addressed your questions. If you have any further concerns, we are appreciated and more than willing to provide any further information or clarification.
>
> If our responses have addressed your concerns, we would appreciate it a lot if you could improve the score.
> Thank you once again for your valuable contribution to improving our work. Looking forward to your feedback.

---

> > ### Author Response · Authors · 2023-11-20
> > **Looking forward to your feedback**
> >
> > Dear Reviewer uEsZ:
> >
> > We apologize for reaching out again during this busy period. But with only three days remaining for the rebuttal, your feedback is crucial for our work. If you have any further concerns, we are appreciated and more than willing to provide any further information or clarification. If our responses have addressed your concerns, we would appreciate it a lot if you could improve the score. Thank you once again for your valuable contribution to improving our work. Looking forward to your feedback.

---

> > > ### Comment · Reviewer_uEsZ · 2023-11-21
> > >
> > > Thanks for your response. It is good to see the results on larger datasets. However, justifications looks not very sound to me. One of the reasons is that different versions of stable diffusion depend on each other, which makes the analysis really confusing. I understand the challenges of training models on LAION-5B dataset. But I think it is necessary to do the ablation study on ImageNet, such as model size, model type, training epochs, and training parameters, etc. Otherwise, the evaluation on these pertained model makes things really ambiguous.
> > >
> > > Moreover, I still think there should be the inspiration for the training strategy is important, as other work (even myself) also found this phenomenon but do not really understand the impact and implications. This paper just verifies the phenomenon without clear implications for practitioners, which limit the significance of the work.

---

> > > > ### Author Response · Authors · 2023-11-22
> > > > **Response to Reviewer uEsZ**
> > > >
> > > > Dear Reviewer uEsZ,
> > > >
> > > > Thank you for your valuable questions. We want to clarify your questions in the following.
> > > >
> > > > **Regarding the ablation study on a large dataset**
> > > >
> > > > 1. During revision, **we have provided extra experiments on ImageNet in Appendix A1 and Figure 9**. This can also be found in **A4** of our response. Specifically, we compared EDM and ADM (which are totally independently trained). We believe the phenomenon still holds in light of Figure 9. (We could not finish these experiments within 12 days of rebuttal, because training on ImageNet is extremely expensive. As a reference, EDM trained their ImageNet models for 13 days using 32 A100 GPUs [1].)
> > > >
> > > > 2. For latent diffusion, as acknowledged by the reviewer, training the latent diffusion model on LAION-5B dataset is very challenging for us. We only use the checkpoints publicly available. Although we agree that independent training would be more convincing, we believe our results still provide some insights. Comparison between V1-3 and V1-4 implies the high reproducibility (0.63) of the two on the same training dataset. In the meanwhile, comparison of V1-1 and V1-2 (or V1-2 and V1-3) suggests that dependencies between those don’t have much effect on the reproducibility (less than 0.21).
> > > >
> > > > **Regarding the implication of our results**
> > > >
> > > > We believe that our results can potentially shed light on improving the efficiency of diffusion models in terms of sampling.
> > > >
> > > > 1. **Sampling efficiency**. For solving inverse problems, recent research [2] shows that utilizing inverse DDIM to transform incomplete measurements into a particular noise in the noise space, and from this noise conditional sampling becomes significantly faster. Our results potentially support this observation by showing that the diffusion model establishes a unique coding between the image and noise spaces. We will add more discussion in the future revision.
> > > >
> > > > 2. **training efficiency**. As for improving training efficiency, once the mapping between image and noise spaces is identified in the generalization regime, we think we might be able to design loss or regularization techniques to improve the diffusion model converging faster to such a unique mapping. This definitely worth further investigation.
> > > >
> > > > Moreover, we believe our result could have potential impacts beyond improving efficiency, as we highlighted in our response **A2** and the discussion in the introduction of our paper.
> > > >
> > > > We hope our responses addressed your questions. If you have any further concerns, we are more than willing to provide any further information or clarification. If our responses have addressed your concerns, we would appreciate it a lot if you could improve the score. Thank you once again for your valuable feedback for improving our work.
> > > >
> > > >
> > > > [1] https://github.com/NVlabs/edm
> > > >
> > > > [2] Liu, Gongye, Haoze Sun, Jiayi Li, Fei Yin, and Yujiu Yang. "Accelerating Diffusion Models for Inverse Problems through Shortcut Sampling." arXiv preprint arXiv:2305.16965 (2023).
> > > >
> > > > [3] Mokady, Ron, Amir Hertz, Kfir Aberman, Yael Pritch, and Daniel Cohen-Or. "Null-text inversion for editing real images using guided diffusion models." In Proceedings of the IEEE/CVF Conference on Computer Vision and Pattern Recognition, pp. 6038-6047. 2023.

---

### Official Review · Reviewer_6ysm · 2023-10-31

**Soundness:** 4 excellent
**Presentation:** 4 excellent
**Contribution:** 4 excellent
**Rating:** 6
**Confidence:** 4

**Summary:**

Diffusion models have demonstrated a strong ability to generate high-quality images. This paper presents a novel discovery referred to as "consistent model reproducibility": regardless of different training configurations, model architectures, and sampling strategies, diffusion models produce nearly identical output content given the same initial noise. The authors theoretically justify the above claim. Besides, they divide model reproducibility manifests in two regimes: memorization regime and generalization regime based on whether the model capacity matches the dataset size. This work provides insights for interpretable and controllable data generation based on diffusion models

**Strengths:**

- The authors both theoretically and empirically show that the same initial noise input results in nearly identical output regardless of the model architecture and training procedure, which is impressive.
- The study is thorough. It covers both unconditional and conditional diffusion models, and considers different sampling strategies, training configurations and model architectures.
- The writing is clear and easy to follow.

**Weaknesses:**

- Lack of discussion about text-to-image diffusion models.
- [minor] "controlable" in the abstract should be controllable.

**Questions:**

Why are the reproducibility scores between transformer-based models and unet-based models low as shown in Figure 7(b)? According to Theorem 1, the reproducibility scores should be high. Could you please provide more explanation?

---

> ### Author Response · Authors · 2023-11-15
> **Response to Reviewer 6ysm**
>
> We thank the reviewer for appreciating the quality of our presentation, experiments, and the timeliness of our results. We value the constructive feedback that the reviewer provided. In the following, we will carefully respond to each of the concerns and suggestions that the reviewer raised.
>
>
>   ___
>
> > **Q1**: Lack of discussion about text-to-image diffusion models.
>
> **A1**:
> We thank the reviewer for the valuable suggestions to study the phenomenon more comprehensively. During the revision, we conducted more experiments considering two extra cases: (i) the Conditional Diffusion Model on the ImageNet Dataset [1], and (ii) Stable Diffusion [2] on the LAION-5B dataset [3]. The results can be found in Appendix A.1 and A.2 of the revised paper. As such, our new experiments not only covered larger datasets in (i, ii), but also text-to-image diffusion models in (ii). Specifically, we find the following:
>
> 1. Conditional Diffusion Model on the ImageNet Dataset: We evaluated the reproducibility of EDM [4] and ADM [5], as shown in the following table:
>
> 	| $\text{RP} \text{ Score}$| EDM  | ADM  |
> 	|----------|------|------|
> 	| **EDM**      | 1.0  | 0.81 |
> 	| **ADM**      | 0.81 | 1.0  |
>
> 	Detailed experimental settings and visual results are presented in Appendix A.1. We find that the model reproducibility still exists on ImageNet.
>
> 2. Stable Diffusion on the LAION-5B Dataset: We evaluated the reproducibility of stable diffusion models with versions v1-1 to v1-4, with their respective reproducibility scores shown in the table below:
>
> 	| $\text{RP} \text{ Score}$ | V1-1 | V1-2 | V1-3 | V1-4 |
> 	|----------|------|------|------|------|
> 	| **V1-1**     | 1.0  | 0.10 | 0.03 | 0.03 |
> 	| **V1-2**     | 0.10 | 1.0  | 0.21 | 0.20 |
> 	| **V1-3**     | 0.03 | 0.21 | 1.0  | **0.63** |
> 	| **V1-4**     | 0.03 | 0.20 | **0.63** | 1.0  |
>
> 	The relationship between V1-1 to V1-4 could be summarized as
> 	- Versions v1-1, v1-2, and v1-3 each are trained on different subsets of the LAION-5B dataset.
> 	- Versions v1-3 and v1-4 share the same training subset from LAION-5B.
> 	- Version v1-2 is resumed from v1-1, while v1-3 and v1-4 are resumed from v1-2.
>
> 	Notably, only versions v1-3 and v1-4 were trained on the exact same dataset, resulting in the highest reproducibility scores (0.63). Lesser but noticeable reproducibility scores (below 0.21) are observed among v1-1, v1-2, and v1-3, this can be attributed to their sequential training and overlapping datasets. There is no model reproducibility between v1-1 and others, because the training datasets are nonoverlapping and different. Further details on the experimental setup and visual results can be found in Appendix A.2.
>
> In conclusion, our findings suggest that this phenomenon persists for larger diffusion models trained on large datasets.
>
> [1]Deng, Jia, Wei Dong, Richard Socher, Li-Jia Li, Kai Li, and Li Fei-Fei. "Imagenet: A large-scale hierarchical image database." In 2009 IEEE conference on computer vision and pattern recognition, pp. 248-255. Ieee, 2009.
>
> [2]Rombach, Robin, Andreas Blattmann, Dominik Lorenz, Patrick Esser, and Björn Ommer. "High-resolution image synthesis with latent diffusion models." In Proceedings of the IEEE/CVF conference on computer vision and pattern recognition, pp. 10684-10695. 2022.
>
> [3] Schuhmann, Christoph, Romain Beaumont, Richard Vencu, Cade Gordon, Ross Wightman, Mehdi Cherti, Theo Coombes et al. "Laion-5b: An open large-scale dataset for training next generation image-text models." Advances in Neural Information Processing Systems 35 (2022): 25278-25294.
>
> [4] Karras, Tero, Miika Aittala, Timo Aila, and Samuli Laine. "Elucidating the design space of diffusion-based generative models." _Advances in Neural Information Processing Systems_ 35 (2022): 26565-26577.
>
> [5] Dhariwal, Prafulla, and Alexander Nichol. "Diffusion models beat gans on image synthesis." _Advances in neural information processing systems_ 34 (2021): 8780-8794.

---

> ### Author Response · Authors · 2023-11-15
> **Response to Reviewer 6ysm**
>
> > **Q2**: Why are the reproducibility scores between transformer-based models and unet-based models low as shown in Figure 7(b)? According to Theorem 1, the reproducibility scores should be high. Could you please provide more explanation?
>
>
> **A2:** Thanks the reviewer for this very good question. When solving inverse problems with diffusion models, our experiments in Figure 7 showed that the model reproducibility between different type of network architectures are very low. We use Diffusion Posterior Sampling (DPS) [1] to solve the inpainting problem, with details provided in Appendix F. We conjecture that the lack of reproducibility across network architectures is due to the following reasons:
> 1. Here, DPS introduces the gradient term $\dfrac{\partial \epsilon_{\theta} ( x_t, t)}{\partial x_t}$ during the sampling, and this extra term might break the reproducibility for different type of architectures. In contrast, for the unconditional diffusion model, the reproducibility holds as long as the denoiser function $\epsilon_{\theta} ( x_t, t)$ is reproducible.
> 2. Second, for image inpainting or inverse problem solving in general, the data $ x_t$ passed into the denoiser  $\epsilon_{\theta}( x_t, t)$ is out-of-distribution. As such, the reproducibility between different types of architectures might not hold for out-of-distribution data generation.
>
> We will incorporate this discussion into the revised paper in Section 4 (when the extra page is provided).
>
> [1] Chung, Hyungjin, Jeongsol Kim, Michael T. Mccann, Marc L. Klasky, and Jong Chul Ye. "Diffusion posterior sampling for general noisy inverse problems." arXiv preprint arXiv:2209.14687 (2022).

---

> > ### Comment · Reviewer_6ysm · 2023-11-22
> > **Thanks for the responses**
> >
> > Thank the authors for the detailed responses.
> >
> > I appreciate the supplemented experiments on text-to-image diffusion models.
> >
> > Yet the explanation for the low scores between diffusion models with different architectures is not convincing. According to theorem 1, the score function also follows the pattern even given out-of-domain data.
> > Why low reproducibility scores between transformer-based and unet-based models might need further exploration.
> >
> > Also, I agree with Reviewer LTJ7 that Theorem 1 might be rather trivial.
> > The model in Theorem 1 is too simple and too ideal: Theorem 1 only considers the optimal score function, which is impossible to obtain in practical training.
> > The result in Eq.(2), which can be derived by computing the minimizer of a quadratic function, is trivial due to the simple modeling.
> >
> > Therefore, I would like to lower my rating to 5.

---

> ### Author Response · Authors · 2023-11-18
> **Looking forward to your feedback**
>
> Dear Reviewer 6ysm:
>
> I would like to begin by expressing my sincere gratitude for the time and effort you have dedicated to reviewing our work. Your insights and feedback are invaluable to us. We hope our responses addressed your questions. If you have any further concerns, we are appreciated and more than willing to provide any further information or clarification.
>
> If our responses have addressed your concerns, we would appreciate it a lot if you could improve the score.
> Thank you once again for your valuable contribution to improving our work. Looking forward to your feedback.

---

> ### Author Response · Authors · 2023-11-22
> **Thanks for your further questions and concerns.**
>
> Dear Reviewer 6ysm,
>
> Thanks for your further questions and concerns.
>
>
> >Q3 ``Also, I agree with Reviewer LTJ7 that Theorem 1 might be rather trivial. The model in Theorem 1 is too simple and too ideal: Theorem 1 only considers the optimal score function, which is impossible to obtain in practical training. The result in Eq.(2), which can be derived by computing the minimizer of a quadratic function, is trivial due to the simple modeling.''
>
> **A3:** We agree with the reviewer that Theorem 1 is simple, but the result is nontrivial. This is corroborated by our experiments in **Figure 5**, where we showed that practical diffusion models do obtain the analytical results **in practice** when the training data size is small. As such, it well explains the reproducibility phenomenon in the **memorization regime**.
>
> Moreover, we want to highlight our results in **Figure 2**, where our work has identified an interesting **generalization regime** where reproducibility and generalizability co-emerge. **The phenomenon in the generalization regimes is very different from that of the memorization regime**, which cannot be simply explained by Reviewer LTJ7’s claim.
>
> >Q4 ``Yet the explanation for the low scores between diffusion models with different architectures is not convincing. According to theorem 1, the score function also follows the pattern even given out-of-domain data. Why low reproducibility scores between transformer-based and unet-based models might need further exploration.''
>
> **A4:** We thank the reviewer for raising the concerns, but we believe that there are some misunderstandings on the phenomenon between the **memorization regime** and **generalization regime** (see Figure 2) that we discovered in our paper.
>
> For inverse problem solving using the diffusion model, the experiment settings are **in the generalization regime**, which is out-of-score of Theorem 1. Our claim on Theorem 1 is only to explain the phenomenon in the memorization regime, as we discussed in A1. We have highlighted this in the figures of our paper.
>
> Moreover, as pointed out by Reviewer vqyB, the low reproducibility between transformer-based and unet-based models for inverse problems in the generalization regime is counter-intuitive and also refutes the claim by Reviewer LTJ7 that this phenomenon is well-known or well-studied. We have provided potential explanations of this in our previous response **A2** that are worth further investigation. Again, this opens a new theoretical question for the diffusion model community.
>
> We hope our responses addressed your questions. If you have any further concerns, we are more than willing to provide any further clarification. Thank you once again for your valuable feedback on improving our work.

---

### Official Review · Reviewer_vqyB · 2023-10-31

**Soundness:** 4 excellent
**Presentation:** 4 excellent
**Contribution:** 4 excellent
**Rating:** 8
**Confidence:** 4

**Summary:**

The paper takes a deep dive into the identifiability of the diffusion model latent space induced by deterministic sampling procedures. In particular, it starts by showcasing the phenomenon that if initialized with the same noisy images and using a deterministic sampler, well-trained diffusion models with widely ranging architectures, training and sampling procedures all produce highly similar images on CIFAR-10. The paper analyses this further and uncovers two distinct regimes where this happens: The overparametrized, ‘memorization regime’, where only a small part of the CIFAR-10 data set is used, and the ‘generalization regime’, where the full CIFAR-10 data set is used. Importantly, there is a gap in between these extremes where the different models do not agree. The effect is measured quantitatively using metrics for comparing image pairs, confirming the result. The mapping from noise to images is visualized and shown again to be similar among different models, as well as smooth. A theoretical study in the ‘memorization’ regime is provided, and reproducibility is demonstrated in conditional generation, inverse problems and fine-tuned models as well.

**Strengths:**

+ The experiments are thorough and demonstrate the phenomenon clearly, and the phenomenon itself is quite striking.
+ The research is well-motivated
+ Although preliminary results on the phenomenon were reported in Song et al (2021), the paper uncovers new effects:
	- The generation is consistent across vastly different architectures, training and sampling procedures
	- The phenomenon occurs in two distinct phases, the ‘memorization’ and ‘generalization’ and regimes, with a clear transition in between where it doesn’t occur.
	- The model reproducibility holds across different types of conditional diffusion models and fine-tuned models as well
+ Potentially the paper opens up more avenues for theoretical work towards understanding diffusion models and how diffusion models generalize, in particular.

References:
Song et al., Score-Based Generative Modeling through Stochastic Differential Equations, 2021, ICLR

**Weaknesses:**

- The present study is mainly on CIFAR-10, and it would be interesting to see how do the results transfer to larger data sets, like ImageNet. Especially when moving to the very large data sets used for text-conditional generation, it doesn't seem obvious that similar results would hold. It would be interesting to see how far can this be extended.

**Questions:**

- When studying the encoding from the noise hyperplane to the image manifold, would it be more appropriate to interpolate the noises using spherical linear interpolation instead of direct linear interpolation? From what I understand, the issue with linear interpolation is that because two randomly sampled high-dimensional noise vectors are, with a high probability, orthogonal to each other, the magnitude of a linearly interpolated vector decreases halfway by a factor of sqrt(2).

Overall, I think that the paper demonstrates and highlights the claimed effects very clearly, and that in addition to the new and more detailed characterization of the phenomenon make the paper worthy of publishing.

---

> ### Author Response · Authors · 2023-11-15
> **Response to Reviewer vqyB**
>
> We very much appreciate the reviewer for acknowledging the quality of our presentation, experiments, and the timeliness of our results. We also thank the reviewer for carefully reading our paper and detailed comments. We value the constructive feedback and encouragement that the reviewer provided. In the following, we carefully respond to each of the concerns and suggestions that the reviewer raised.
>
> ___
>
> > **Q1**: “The present study is mainly on CIFAR-10, and it would be interesting to see how do the results transfer to larger data sets, like ImageNet. Especially when moving to the very large data sets used for text-conditional generation.”
>
> **A1**:
> We thank the reviewer for the valuable suggestions to study the phenomenon more comprehensively. During the revision, we conducted more experiments considering two extra cases: (i) the Conditional Diffusion Model on the ImageNet Dataset [1], and (ii) Stable Diffusion [2] on the LAION-5B dataset [3]. The results can be found in Appendix A.1 and A.2 of the revised paper. As such, our new experiments not only covered larger datasets in (i, ii), but also text-to-image diffusion models in (ii). Specifically, we find the following:
>
> 1. Conditional Diffusion Model on the ImageNet Dataset: We evaluated the reproducibility of EDM [4] and ADM [5], as shown in the following table:
>
> 	|$\text{RP} \text{Score}$|EDM|ADM|
> 	|------|------|------|
> 	| **EDM**      | 1.0  | 0.81 |
> 	| **ADM**      | 0.81 | 1.0  |
>
> 	Detailed experimental settings and visual results are presented in Appendix A.1. We find that the model reproducibility still exists on ImageNet.
>
> 2. Stable Diffusion on the LAION-5B Dataset: We evaluated the reproducibility of stable diffusion models with versions v1-1 to v1-4, with their respective reproducibility scores shown in the table below:
>
> 	| $\text{RP} \text{ Score}$ | V1-1 | V1-2 | V1-3 | V1-4 |
> 	|----------|------|------|------|------|
> 	| **V1-1**     | 1.0  | 0.10 | 0.03 | 0.03 |
> 	| **V1-2**     | 0.10 | 1.0  | 0.21 | 0.20 |
> 	| **V1-3**     | 0.03 | 0.21 | 1.0  | **0.63** |
> 	| **V1-4**     | 0.03 | 0.20 | **0.63** | 1.0  |
>
> 	The relationship between V1-1 to V1-4 could be summarized as
> 	- Versions v1-1, v1-2, and v1-3 each are trained on different subsets of the LAION-5B dataset.
> 	- Versions v1-3 and v1-4 share the same training subset from LAION-5B.
> 	- Version v1-2 is resumed from v1-1, while v1-3 and v1-4 are resumed from v1-2.
>
> 	Notably, only versions v1-3 and v1-4 were trained on the exact same dataset, resulting in the highest reproducibility scores (0.63). Lesser but noticeable reproducibility scores (below 0.21) are observed among v1-1, v1-2, and v1-3, this can be attributed to their sequential training and overlapping datasets. There is no model reproducibility between v1-1 and others, because the training datasets are nonoverlapping and different. Further details on the experimental setup and visual results can be found in Appendix  A.2.
>
> In conclusion, our findings suggest that this phenomenon persists for larger diffusion models trained on large datasets.
>
> [1]Deng, Jia, Wei Dong, Richard Socher, Li-Jia Li, Kai Li, and Li Fei-Fei. "Imagenet: A large-scale hierarchical image database." In 2009 IEEE conference on computer vision and pattern recognition, pp. 248-255. Ieee, 2009.
>
> [2]Rombach, Robin, Andreas Blattmann, Dominik Lorenz, Patrick Esser, and Björn Ommer. "High-resolution image synthesis with latent diffusion models." In Proceedings of the IEEE/CVF conference on computer vision and pattern recognition, pp. 10684-10695. 2022.
>
> [3] Schuhmann, Christoph, Romain Beaumont, Richard Vencu, Cade Gordon, Ross Wightman, Mehdi Cherti, Theo Coombes et al. "Laion-5b: An open large-scale dataset for training next generation image-text models." Advances in Neural Information Processing Systems 35 (2022): 25278-25294.
>
> [4] Karras, Tero, Miika Aittala, Timo Aila, and Samuli Laine. "Elucidating the design space of diffusion-based generative models." _Advances in Neural Information Processing Systems_ 35 (2022): 26565-26577.
>
> [5] Dhariwal, Prafulla, and Alexander Nichol. "Diffusion models beat gans on image synthesis." _Advances in neural information processing systems_ 34 (2021): 8780-8794.

---

> ### Author Response · Authors · 2023-11-15
> **Response to Reviewer vqyB**
>
> > **Q2**: ”When studying the encoding from the noise hyperplane to the image manifold, would it be more appropriate to interpolate the noises using spherical linear interpolation instead of direct linear interpolation?”
>
>  **A2**:
> Thanks for your very insightful suggestion. Given the spherical linear interpolation is confined to a single dimension and time limit, we opted not to visualize the manifold reproducibility in a figure. Instead, we developed a manifold reproducibility score to measure the manifold reproducibility. The metric is defined as follows:
>
> The metric begins by selecting two initial noise vectors, $(\mathbf{\epsilon}^{(0)}, \mathbf{\epsilon}^{(1)})$, from the noise space $\mathcal E$. These vectors are then processed through two distinct diffusion model architectures, resulting in pairs of clear images: ($\mathbf x^{(0)}_1$, $\mathbf x^{(1)}_1$) from the first model and ($\mathbf x^{(0)}_2$, $\mathbf x^{(1)}_2$) from the second. The manifold reproducibility score for the two unconditional diffusion models is defined as:
>
>
> $RP_{manifold} Score :=  \mathbb{P}( \mathcal M_{SSCD} ( x^{(\alpha)}_1,  x^{(\alpha)}_2)>0.6 \mid \alpha \in [0,1] )$
>
> where $\mathbf x^{(\alpha)}_i$ are generated from spherical linear interpolation based on ($\mathbf x^{(0)}_i$, $\mathbf x^{(1)}_i$):
> $$
> \begin{align*}
>     & \mathbf x^{(\alpha)}_i = \frac{\text{sin}((1 - \alpha) \theta)}{\text{sin}(\theta) } \mathbf x^{(0)}_i + \frac{\text{sin}(\alpha \theta)}{\theta} \mathbf x^{(1)}_i, \ \ i \in \{1, 2\} \\
>     & \theta = \text{arccos} \left(\frac{ \left(x^{(0)}_i\right )^T x^{(1)}_i}{||x^{(0)}_i|| ||x^{(1)}_i||} \right)
> \end{align*}
> $$
>
> We applied this score to compare the manifold reproducibility across various unconditional diffusion models.  Results are shown below. Detailed descriptions of the experimental settings can be found in Appendix A.3. In this case, our observation of model reproducibility is still consistent with our current findings.
>
> | $\text{RP}_{manifold} \text{ Score}$   | DDPMv6 | EDMv1 | Multistagev1 |
> |--------------|--------|-------|--------------|
> | **DDPMv6**       | 1.0    | 0.90  | 1.0          |
> | **EDMv1**        | 0.90   | 1.0   | 0.93         |
> | **Multistagev1** | 1.0    | 0.93  | 1.0          |

---

> > ### Author Response · Authors · 2023-11-18
> > **Looking forward to your feedback.**
> >
> > Dear Reviewer vqyB:
> >
> > I would like to begin by expressing my sincere gratitude for the time and effort you have dedicated to reviewing our work. Your insights and feedback are invaluable to us. We hope our responses addressed your questions. If you have any further concerns,  we are appreciated and more than willing to provide any further information or clarification.
> >
> > Thank you once again for your valuable contribution to improving our work. Looking forward to your feedback.

---

> > > ### Comment · Reviewer_vqyB · 2023-11-21
> > > **Response to rebuttal**
> > >
> > > I thank the authors for the new results! I agree that the new results do make the paper more convincing than before. I will not change the score for now, but will consider it after discussion with the reviewers.

---

> > > > ### Author Response · Authors · 2023-11-22
> > > > **Thanks for your review**
> > > >
> > > > Dear Reviewer vqyB,
> > > >
> > > > We wish to extend our gratitude for your responsible and high-quality review of our paper. Your insightful summary precisely captures the core idea of our result, and your high-quality feedback is invaluable for improving our work. Reviewers like you play a crucial role in fostering a better research community and in nurturing high-quality works. You have set a very high standard for top machine learning reviewers.

---

### Official Review · Reviewer_LTJ7 · 2023-11-05

**Soundness:** 3 good
**Presentation:** 3 good
**Contribution:** 1 poor
**Rating:** 3
**Confidence:** 3

**Summary:**

The authors observe that different trained diffusion models map to almost identical output images when the deterministic ODE sampling starts from the same initial noise. They validate this experiment with different models and they measure that the behavior is consistent for models that are trained separately, with different architectures, different samplers, and even different perturbation kernels. The authors provide a theoretical justification for this phenomenon in a toy setting (with infinite model capacity and a target distribution of many diracs).

**Strengths:**

- The paper is easy to follow.
- The authors conduct numerous experiments to verify their findings.
- Understanding the behavior of deterministic samplers is an interesting and timely research topic.

**Weaknesses:**

I believe the main finding of this paper is already known. The Probability Flow ODE depends on the functions $f$, $g$, and the score. For given functions $f$, $g$, the score function is unique! There is a unique score function because there is a unique likelihood function induced by corrupting the data distribution. All diffusion models are trained to estimate the exact same score function — even if they have different architectures and are trained separately — the target is always the same. Hence, if the diffusion models are trained perfectly, then it is actually expected that they will all map the same noise to the same output. The small deviations in the shown images in Figure 1, just indicate that there are some learning errors. The finding should stay the same even if we use different samplers, as long as the samplers arrive with guarantees that given enough steps and access to the score they will sample from the right distribution. The only surprising fact to me in this paper is the claim that models trained with different corruption processes will have this property. This doesn’t make much sense to me because for VE SDE and VP SDE for example, even the terminal distribution is different, so how do we even start from the same noise? What distribution does this noise follow?

**Questions:**

There is a chance that I am missing something fundamental about this paper. I would really like the authors to clarify, if possible, why their main finding is different compared to prior work.

---

> ### Author Response · Authors · 2023-11-15
> **Response to Reviewer LTJ7**
>
> We thank the reviewer for acknowledging the quality of our presentation, experiments, and the timeliness of our results. We value the constructive feedback that the reviewer provided. Although we agree that the some preliminary observations on the reproducibility phenomenon were reported by Song et al (2021) [1] in a very empirical way, our investigation delves deeper and provides a more systematic study about the fundamentals for this phenomenon for generic diffusion models which haven’t been done in previous works to our best knowledge, offering a much more thorough analysis. In particular, the key contributions/findings of our paper that differ from prior works include:
>
> -   we introduced quantitative measures to study reproducibility more systematically;
>
> -   we demonstrate that reproducibility manifests in two unique regimes and is closely tied to the generalizability of diffusion models, as depicted in Figure 2 of our study—an interesting relationship that has never been explored previously.
>
> -   we also studied model reproducibility in conditional diffusion models, diffusion models for inverse problems, and fine-tuning, which indicates how this phenomenon could be useful in these different applications.
>
>
> Reviewer vqyB has also noted these significant findings in his comments. In the following, we carefully respond to each of the concerns that the reviewer raised.
> ___
>
> > **Q1**:   “I believe the main finding of this paper is already known.” “why their main finding is different compared to prior work.”
>
> **A1**:
> We agree with the reviewer that Song et al. (2021) presented initial findings on the reproducibility phenomenon with a brief description of this observation. Please note that we've also acknowledged this point at the bottom of Page 3 of our paper. However, we respectfully disagree that our main discovery is well-known or extensively researched. This has also been pointed out by Reviewer vqyB.
>
> 1. First, in Section 2, we introduced reproducibility metrics to study this phenomenon more systematically and quantitatively. Our findings that reproducibility appears across different noise corruption processes have not been shown in prior works and the phenomenon is nontrivial, which we will discuss in detail in A2 for Q2.
>
> 2. Second, in Section 3, our work showed that model reproducibility occurs in two distinct regimes, the “memorization regime” and “generalizability regime”, and model reproducibility has a strong correlation with model generalizability (see Figure 2). This phenomenon has never been discovered nor studied in previous works, and we believe this is very important for understanding the generalizability properties of diffusion models. Moreover, we have theoretically and experimentally justified the reproducibility in the memorization regime, as illustrated in Figure 5.
>
> 3. Third, in Section 4 we studied the model reproducibility of diffusion models under different settings, such as conditional diffusion models, diffusion models for inverse problems, and fine-tuning diffusion models. For each of these different settings, the model reproducibility manifests in different but principled ways. These have never been studied in previous works.
>
> [1] Song, Yang, Jascha Sohl-Dickstein, Diederik P. Kingma, Abhishek Kumar, Stefano Ermon, and Ben Poole. "Score-based generative modeling through stochastic differential equations." _arXiv preprint arXiv:2011.13456_ (2020).

---

> ### Author Response · Authors · 2023-11-15
> **Response to Reviewer LTJ7**
>
> > **Q2**:  "The Probability Flow ODE depends on the functions f, g, and the score. For given functions f, g, the score function is unique! There is a unique score function because there is a unique likelihood function induced by corrupting the data distribution. All diffusion models are trained to estimate the exact same score function — even if they have different architectures and are trained separately — the target is always the same. Hence, if the diffusion models are trained perfectly, then it is expected that they will all map the same noise to the same output.”
>
> **A2**:
> First of all, we appreciate the reviewer's thoughtful remarks regarding the model reproducibility. However, to the best of our knowledge, except for the preliminary findings reported by Song et al. in 2021 [1], we have not come across any empirical or theoretical evidence substantiating the above claims that (i) “all diffusion models (in terms of all different settings and cases) are designed to estimate exactly the same score function in practice”, and (ii) “all diffusion models (in terms of all different settings and cases) can always be perfectly trained for score estimation”. We would appreciate it if the reviewer could share the references that can support these claims other than Song et al. 2021.
>
> On the other hand, to partially support/refute the reviewer’s claims, our work has quantitatively conducted extensive experiments to study this problem in both memorization and generalization regimes under various settings, and we provided a theoretical study in our work in the memorization regime.
>
> 1.  First, in the memorization regime when the training data is limited, we demonstrate that the diffusion model can only reproduce or “memorize” training data. And the reviewer’s Claims (i) and (ii) are corroborated by both our theoretical analysis and experimental findings. Our research demonstrates the existence of an optimal denoiser and scoring function, as established in our Theorem 1. Furthermore, our work showed that the practical diffusion models converge towards this optimal denoiser, which is unique given f and g, as evidenced in Figure 5. Therefore, in the memorization regime, this observation is in agreement with the reviewer's statement.
>
> 2.  In the generalization regime, where the model reproducibility and generalizability co-exist, whether and how the diffusion model has been trained perfectly is not clear. As depicted in Figure 13 of our revised paper, it is evident that in the generalization regime, the denoiser obtains a relatively high training loss, but still has good generations, good generalizability, and also has reproducibility. This requires additional investigation before we can assert this claim, and our empirical results have raised numerous intriguing theoretical questions that merit further exploration.
>
> 3.  Moreover, our results surprisingly show that model reproducibility still exists in the generalization regime, even if we use different functions $f$,$g$ by using different perturbation kernels (we address the noise distribution question in Q3). This observation may diverge from the assumption of the reviewer’s claim.
>
> [1] Song, Yang, Jascha Sohl-Dickstein, Diederik P. Kingma, Abhishek Kumar, Stefano Ermon, and Ben Poole. "Score-based generative modeling through stochastic differential equations." ICLR, 2021.

---

> ### Author Response · Authors · 2023-11-15
> **Response to Reviewer LTJ7**
>
> > **Q3**:  “The claim, that models trained with different corruption processes will have this property, doesn’t make much sense to me. This is because for VE SDE and VP SDE for example, even the terminal distribution is different, so how do we even start from the same noise? What distribution does this noise follow?”
>
> **A3**:
> Although  using different perturbation kernels will result in different noise distributions, the distribution difference only lies in the scaling of the variance for the Gaussian. Therefore, for a fair comparison, we can always get rid of the scaling issues by normalizing the variance and then sampling from the noise.  We have discussed this in detail in “Initial Noise Consistency” of Appendix B of our work.
>
> For example, for VP and subVP noise perturbation kernels in [1], we define the noise space as $\mathcal E = \mathcal{N}(\mathbf 0, \mathbf I)$, whereas the VE noise perturbation kernel [1] introduces a distinct noise space with $\mathcal E = \mathcal{N}(\mathbf 0,  \sigma^2_{\text{max}} \cdot \mathbf I)$, where $\sigma_{\text{max}}$ is predefined. So during the experiment, we sample 10K initial noise $\mathbf \epsilon_{\text{vp, subvp}} \sim \mathcal{N}(\mathbf 0, \mathbf I)$ for the generation with VP and subVP noise perturbation kernel. For VE noise perturbation kernel, the initial noise is scaled as $\mathbf \epsilon_{\text{ve}} = \sigma_{\text{max}} \mathbf \epsilon_{\text{vp, subvp}}$.
>
> [1] Song, Yang, Jascha Sohl-Dickstein, Diederik P. Kingma, Abhishek Kumar, Stefano Ermon, and Ben Poole. "Score-based generative modeling through stochastic differential equations." ICLR, 2021.

---

> ### Author Response · Authors · 2023-11-18
> **Looking forward to your feedback**
>
> Dear Reviewer LTJ7:
>
> I would like to begin by expressing my sincere gratitude for the time and effort you have dedicated to reviewing our work. Your insights and feedback are invaluable to us. We hope our responses addressed your questions. If you have any further concerns,  we are appreciated and more than willing to provide any further information or clarification.
>
> If our responses have addressed your concerns, we would appreciate it a lot if you could improve the score.
> Thank you once again for your valuable contribution to improving our work. Looking forward to your feedback.

---

> > ### Author Response · Authors · 2023-11-20
> > **Looking forward to your feedback**
> >
> > Dear Reviewer LTJ7:
> >
> > We apologize for reaching out again during this busy period. But with only three days remaining for the rebuttal, your feedback is crucial for our work. If you have any further concerns, we are appreciated and more than willing to provide any further information or clarification. If our responses have addressed your concerns, we would appreciate it a lot if you could improve the score. Thank you once again for your valuable contribution to improving our work. Looking forward to your feedback.

---

### Author Response · Authors · 2023-11-15
**Summary of updates**

We express our gratitude to all reviewers for their insightful feedback and valuable suggestions, which have significantly contributed to the improvement of our paper.

In summary, our work provided a comprehensive study of model reproducibility of diffusion models, and more importantly, showed that it manifests in two different learning regimes which are strongly related to model generalization.

Most of the reviewers find our study interesting (all reviewers), well-motivated (Reviewer vqyB), comprehensive (Reviewer LTJ7, Reviewer vqyB, Reviewer 6ysm), and well-represented (Reviewer LTJ7, Reviewer vqyB, Reviewer 6ysm).

In response to their comments, we have carefully revised our paper and provided a detailed summary of these revisions in Appendix A. All the revisions of our paper are marked in blue color.

In the revision, we addressed the major concerns of most reviewers on **a study of larger datasets and text-to-image diffusion models**. Specifically, in Appendix A.1, we now include results using ImageNet [1]. In Appendix A.2, we have incorporated an examination of the text-to-image diffusion model, particularly focusing on stable diffusion [2]. Our results show that the model reproducibility persists in these settings.

Additionally, we address individual questions from each reviewer as follows: (i) we address Reviewer vqyB’s question on spherical linear interpolation in Appendix A.3, (ii) we address Reviewer uEsZ’s question on the impact of the number of function evaluations (NFE) in Appendix A.4, and (iii) in response to Reviewer LTJ7's comments on perfect score function learning, we conduct more experiments in Appendix A.5, which complements our results in Figure 5.

[1] Deng, Jia, Wei Dong, Richard Socher, Li-Jia Li, Kai Li, and Li Fei-Fei. "Imagenet: A large-scale hierarchical image database." In 2009 IEEE conference on computer vision and pattern recognition, pp. 248-255. Ieee, 2009.

[2] Rombach, Robin, Andreas Blattmann, Dominik Lorenz, Patrick Esser, and Björn Ommer. "High-resolution image synthesis with latent diffusion models." In Proceedings of the IEEE/CVF conference on computer vision and pattern recognition, pp. 10684-10695. 2022.

---

### Comment · Area_Chair_s66y · 2023-11-21
**Discussing the Significance of Empirical Findings**

There is evident disagreement among the reviewers regarding the significance of the empirical findings related to the reproducibility and consistency in diffusion models. One primary criticism is that these empirical findings are already known, and therefore, they do not provide sufficient grounds for acceptance into ICLR. Another major concern is that these empirical findings do not (yet) yield improvement in training.

Reviewers LTJ7 and uEsZ: Please review the authors' rebuttal to determine if your concerns persist.

AC's comments: I commend the thorough empirical research on reproducibility and consistency in diffusion modeling, which I believe will be valuable to the community. However, it is worth noting that these findings, if not already familiar to the community, have been previously alluded to in various sources, including but not limited to Song et al. (2021).

One approach to contextualize the empirical results is through the perspective of denoising score matching, interpreted via Tweedie's formula. This suggests that a central goal in diffusion modeling is to estimate $E[ x_0 | x_t] $, essentially the mean of all clean images that could lead to $ x_t $ after the addition of Gaussian noise at level $ \sigma_t $. This concept has been discussed in several papers, including:

- Luo, C. (2022) in 'Understanding diffusion models: A unified perspective' ([arXiv:2208.11970](https://arxiv.org/abs/2208.11970)).

- Chung, H., et al. (2022) in 'Improving diffusion models for inverse problems using manifold constraints' (Advances in Neural Information Processing Systems, 35, 25683-25696).

- Kim, D., et al. (2023) in 'Consistency Trajectory Models: Learning Probability Flow ODE Trajectory of Diffusion' ([arXiv preprint arXiv:2310.02279](https://arxiv.org/abs/2310.02279)).

Additionally, interpreting the denoising L2 loss in diffusion modeling as a Bregman divergence also leads to a similar conclusion about diffusion modeling's goal to estimate $E[ x_0 | x_t] $. Pertinent literature on Bregman divergence and its connection to diffusion modeling includes:

- Banerjee, A., et al. (2005) in 'Clustering with Bregman Divergences' (Journal of Machine Learning Research, 6, 1705-1749).
- Zhou, M., et al. (2023) in 'Beta Diffusion' ([arXiv preprint arXiv:2309.07867](https://arxiv.org/abs/2309.07867)).

Given this understanding, I would anticipate that using the same ODE solver starting from the same random noise should produce similar images, even when employed with different models trained on the same dataset. In fact, I would also expect the SDE/DDPM solver to generate similar images if the noise injections remain consistent at each stochastic reverse diffusion step.

I eagerly anticipate responses from both the authors and reviewers.

---

> ### Comment · Reviewer_vqyB · 2023-11-21
> **On whether we expect similar images in the "generalization regime"**
>
> I would like to briefly here also raise discussion on how obvious is the result that all the different models produce similar images. From what I can see, it is true that the goal in diffusion models is to estimate the $\mathbb{E}[x_0|x_t]$, and if all the models accurately estimate this (and thus the ground-truth score function for the data set), it seems clear to me that all models behave similarly: They reproduce the original data set. This is also pointed out in Theorem 1 of the paper, regarding the "memorization regime". I may be mistaken, but I do not see a similar result holding in the "generalization regime", where none of the models recover the ground-truth score, but instead make "errors" and thus generalize. Intuitively, it seems that all models make highly similar errors for some reason or the other. This seems quite non-obvious to me, and in that sense an interesting direction for research.
>
> Additionally, from what I can see (but again may be wrong), the paper contains new aspects of the phenomenon that have not been noticed in previous work. For instance,  1) The fact that there's a point between the "generalization" and "memorization" regimes where the models do not produce similar results. 2) The paper points out a case where reproducibility does not hold (UViT/DiT compared to DDPM models in inpainting).
>
> I wonder if the others would agree on these points?

---

> ### Author Response · Authors · 2023-11-23
> **Discussion from the authors (part 1)**
>
> Dear AC and reviewers:
>
> We really appreciate it that AC initiated such a discussion, and thank Reviewer vqyB for helping us to clarify the contributions of our work. We thank Reviewer LTJ7, 6ysm, and uEsZ for their valuable feedback for improving our work. We also want to thank the AC for pointing us to valuable references, which we will cite and carefully discuss in our revised paper.
>
> In the following, we want to use this opportunity to further clarify the contributions of our work.
>
> 1. **We showed model reproducibility in two distinct “memorization’’ and “generalization” regimes, as highlighted in Figure 2.** This finding in Figure 2 and Section 3 is significant because it demonstrates that reproducibility in diffusion models is not just a byproduct of a deterministic sampling process. Instead, it arises from various factors like dataset size, model capacity, and generalization ability. We found that model reproducibility occurs either in the memorization regime when the dataset is small enough for the model to memorize training data (as seen in the left part of Figure 2(a)(b)), or in the generalization regime when the dataset is large enough for the model to generate new samples (right part of Figure 2(a)(b)).  Moreover, Figure 2 shows a co-emergence of reproducibility and generalizability, which we believe is quite intriguing.  Prior research often mixed up these two scenarios without distinguishing them. Our work uniquely identifies the root of this phenomenon, showing that reproducibility stems from the model either memorizing data or learning the underlying data distribution (generalization), a topic not thoroughly examined in previous studies.
> 2. **Diffusion models estimate different distributions in these two distinct regimes while maintaining reproducibility.** The AC made an insightful observation that the primary objective in diffusion models is to estimate $\mathbb E[x_0|x_t]$. However, the practical methodologies by which diffusion models are trained for this estimation have not been thoroughly explored in previous studies. Our research offers a comprehensive investigation of this aspect.
>    - In the memorization regime, we demonstrate both theoretically and experimentally that the diffusion model has the capacity to effectively reduce the training loss as defined in equation (1), which can be depicted in the left portion of Figure 13. This process leads to convergence towards the optimal denoiser in equation (2) of Theorem 1, as illustrated in the left portion of Figure 5. The mathematical equivalence of this optimal denoiser to $\mathbb E_{x_0 \sim p_{\text{training data}}}[x_0|x_t]$ is established in [1], where $p_{\text{training data}}$ represents the distribution of the training data.
>    - In the context of generalization, the findings presented in Figure 13 indicate that the diffusion model does not appear to estimate $\mathbb E_{x_0 \sim p_{\text{training data}}}[x_0|x_t]$. This is obvious from the substantial increase in the training loss (equation (1)) as the size of the training data increases. We posit that, in the generalization regime, the diffusion model may instead be learning to estimate $\mathbb E_{x_0 \sim p_{\text{true data}}}[x_0|x_t]$, where $p_{\text{true data}}$ represents the distribution of the true underlying data. This hypothesis can be supported by large generalization score in the right part of Figure 2(b), worth further theoretical investigation.
>
> 3. **Systematic yet counter-intuitive findings in the generalization regime for further investigation.** We highlight these findings below:
>    - **Model reproducibility holds even under different perturbation processes.** This implies that in the generalization regime, the diffusion model has the ability to consistently converge and learn the actual data distribution, denoted as $p_{\text{training data}}$, regardless of the specific noise perturbation process applied.
>    - **Model reproducibility diminishes between different types of network architectures for inverse problems.** Specifically, in the context of inverse problem solving, we observe that reproducibility does not persist when comparing U-Net-based and Transformer-based architectures in the generalization regime. This implies that by adding constraints or conditions, the posterior distribution may change in a way that it does not inherit the reproducibility from unconditional data distribution. These findings may be potentially quite insightful to guide how we utilize the distribution prior learned from diffusion models for solving inverse problems,  which is unexplored and worth further investigation.
>    - In the transition between memorization and generalization regimes, there is no reproducibility between different diffusion models. There is an interesting phase transition phenomenon to be well understood.
>
>
> [1] Yi, Mingyang, Jiacheng Sun, and Zhenguo Li. "On the Generalization of Diffusion Model." arXiv preprint arXiv:2305.14712 (2023).

---

> ### Author Response · Authors · 2023-11-23
> **Discussion from the authors (part 2)**
>
> 4. **Model reproducibility beyond unconditional diffusion models.** Our study has revealed a more structured pattern of reproducibility when examining conditional diffusion models in the generalization regime. This applies to scenarios like diffusion models for inverse problems and fine-tuning diffusion models. Although these settings have distinct characteristics compared to the unconditional diffusion model, they exhibit a systematic and well-defined behavior. As illustrated in Figure 6, there is a distinctive reproducibility pattern that remains consistent between conditional and unconditional diffusion models. To our knowledge, these findings represent novel and inspiring contributions to the existing body of research.
>
> In summary, while the idea may appear intuitive initially, the reproducibility of diffusion models remains a largely uncharted territory that worth a more comprehensive investigation by the research community, rather than relying solely on intuition. Our study represents one of the first efforts in this area. We are confident that our result is new and inspiring, particularly in the generalization regime as illustrated in Figure 2.
>
> Moreover, we believe that our paper opened many interesting theoretical problems to the field on ``why diffusion model generalizes’’, that worth further investigation, and our result sheds insights for more interpretable and controllable data generation in practice.

---

### Meta-Review · Area_Chair_s66y · 2023-12-08

**Metareview:**

The reviewers acknowledge the significance of the observations on reproducibility and consistency in diffusion models presented in the paper. However, they highlight that these findings align with those of prior studies. The paper is criticized for not providing clear evidence on how these observations can lead to meaningful improvements in training or sampling. This absence raises concerns regarding the practical impact and implementation of the insights. Consequently, these considerations contribute to a weak rejection of the paper.

**Justification For Why Not Higher Score:**

Most reviewers express concerns about the originality of the insights presented in the paper and question their practical impact.

**Justification For Why Not Lower Score:**

N/A

---

### Decision · Program_Chairs · 2024-01-16

Reject